# Neural dynamics of visual ambiguity resolution by perceptual prior

Matthew W Flounders[1], Carlos González-García[2], Richard Hardstone[1], Biyu J He[1,3,4,5]*

[1]Neuroscience Institute, New York University Langone Medical Center, New York, United States; [2]Department of Experimental Psychology, Ghent University, Ghent, Belgium; [3]Department of Neurology, New York University Langone Medical Center, New York, United States; [4]Department of Neuroscience and Physiology, New York University Langone Medical Center, New York, United States; [5]Department of Radiology, New York University Langone Medical Center, New York, United States

**Abstract** Past experiences have enormous power in shaping our daily perception. Currently, dynamical neural mechanisms underlying this process remain mysterious. Exploiting a dramatic visual phenomenon, where a single experience of viewing a clear image allows instant recognition of a related degraded image, we investigated this question using MEG and 7 Tesla fMRI in humans. We observed that following the acquisition of perceptual priors, different degraded images are represented much more distinctly in neural dynamics starting from ~500 ms after stimulus onset. Content-specific neural activity related to stimulus-feature processing dominated within 300 ms after stimulus onset, while content-specific neural activity related to recognition processing dominated from 500 ms onward. Model-driven MEG-fMRI data fusion revealed the spatiotemporal evolution of neural activities involved in stimulus, attentional, and recognition processing. Together, these findings shed light on how experience shapes perceptual processing across space and time in the brain.
DOI: https://doi.org/10.7554/eLife.41861.001

*For correspondence:
biyu.jade.he@gmail.com

Competing interests: The authors declare that no competing interests exist.

## Introduction

Perception reflects not only immediate patterns of sensory inputs but also memories acquired through prior experiences with the world (*Helmholtz, 1924*; *Albright, 2012*). For instance, reading handwriting greatly depends on our existing knowledge of vocabulary and grammar. Stored representations from previous experiences provide likely interpretations of sensory data about their cause and meaning, overcoming the ever-present noise, ambiguity and incompleteness of retinal image. However, to date neural mechanisms underlying prior experiences' influence on visual perception remain largely unknown.

Here, we adopted the Mooney image paradigm to investigate visual perception of identical sensory input that results in distinct perceptual outcomes depending on whether or not prior knowledge is present. These Mooney images are degraded, two-tone images created from natural photographs of objects and animals. Even after multiple presentations, the content of these images typically remains unrecognizable. However, once exposed to the corresponding non-degraded original photograph, subjects effortlessly recognize the Mooney image in future presentations – a disambiguation effect that lasts for days, months, even a lifetime (*Ludmer et al., 2011*; *Albright, 2012*). This phenomenon illustrates that a prior that guides perception can be established in a remarkably fast and robust manner. Thus, Mooney images offer an experimentally controlled paradigm for dissecting how prior experience shapes perceptual processing.

Previous neuroimaging studies have observed that disambiguation of Mooney images induces widespread activation and enhanced image-specific information in both visual and frontoparietal cortices (*Dolan et al., 1997*; *Hegdé and Kersten, 2010*; *Hsieh et al., 2010*; *Gorlin et al., 2012*; *van Loon et al., 2016*; *González-García et al., 2018*). However, due to the slow temporal resolution of these techniques (positron emission tomography and fMRI), the temporal dynamics underlying this effect remains unknown. This is an important open question because behavioral studies have shown that recognition of Mooney images, even after disambiguation, is slow – with reaction times at around 1.2 s (*Hegdé and Kersten, 2010*). By contrast, neural dynamics underlying recognition of intact, unambiguous images, as well as scene-facilitation of object recognition, typically conclude within 500 ms (*Carlson et al., 2013*; *van de Nieuwenhuijzen et al., 2013*; *Kaiser et al., 2016*; *Brandman and Peelen, 2017*). Together with a recent finding of altered content-specific neural representations in frontoparietal regions following Mooney image disambiguation (*González-García et al., 2018*), these observations raise the intriguing possibility that slow (taking longer than 500 ms), long-distance recurrent neural dynamics involving large-scale brain networks are necessary for prior-experience-guided visual recognition.

Several previous EEG and MEG studies reported disambiguation-induced decrease in beta-band power and increase in gamma-band power (*Grützner et al., 2010*; *Minami et al., 2014*; *Moratti et al., 2014*), but these effects could potentially be attributed to non-content-specific effects such as increased attention, salience or decreased task difficulty following disambiguation. To unravel neural mechanisms underlying prior experience's influence on perception, an important unanswered question is how different information processing stages are dynamically encoded in neural activities.

Here, we probe the dynamical encoding of perceptual state (before or after disambiguation) as well as the physical features and recognition outcomes related to individual images in neural activities, using multivariate pattern decoding and representational similarity analysis (RSA) applied to whole-head MEG data. In addition, to illuminate the anatomical distribution of the evolving neural dynamics involved in different information processing stages, we applied model-driven cross-modal RSA (*Hebart et al., 2018*) to combine the high temporal resolution of MEG data with high spatial resolution of 7T fMRI data collected using a similar paradigm. These approaches allowed us to spatiotemporally resolve neural dynamics underlying different stages of information processing during prior-guided visual perception.

## Results

### Paradigm and behavioral results

Eighteen subjects were shown 33 Mooney images containing animals or manmade objects. Each Mooney image was presented six times before its corresponding grayscale image was shown to the subject, and six times after. Following each Mooney image presentation, subjects reported whether they could recognize the image using a button press ('subjective recognition'). Each MEG 'run' (*Figure 1*; for details see Materials and methods, *Task paradigm*) included three different grayscale images, their corresponding post-disambiguation Mooney images, and three new Mooney images shown pre-disambiguation (their corresponding grayscale images would be shown in the next run). To ensure that subjects' self-reported recognition matched the true content of the Mooney images, at the end of each run, Mooney images presented during that run were shown again and participants were asked to verbally report what they saw in the image and were allowed to answer 'unknown'. This resulted in a verbal test for each Mooney image once before disambiguation and once after disambiguation ('verbal identification'). Verbal responses were scored as correct or incorrect using a pre-determined list of acceptable responses for each image. In addition, six Mooney images were presented with non-matching grayscale images using identical block and run structure, which served as controls for the effect of repetition ('catch image sets', as opposed to the 33 'real image sets' described earlier).

Viewing the corresponding grayscale image had a dramatic effect on Mooney image recognition as shown by the following results. First, we compared recognition rates between pre- and post-disambiguation stages using a two-way repeated-measures ANOVA [factors: presentation stage and image-set type (real vs. catch)]. This analysis was carried out separately using subjective recognition

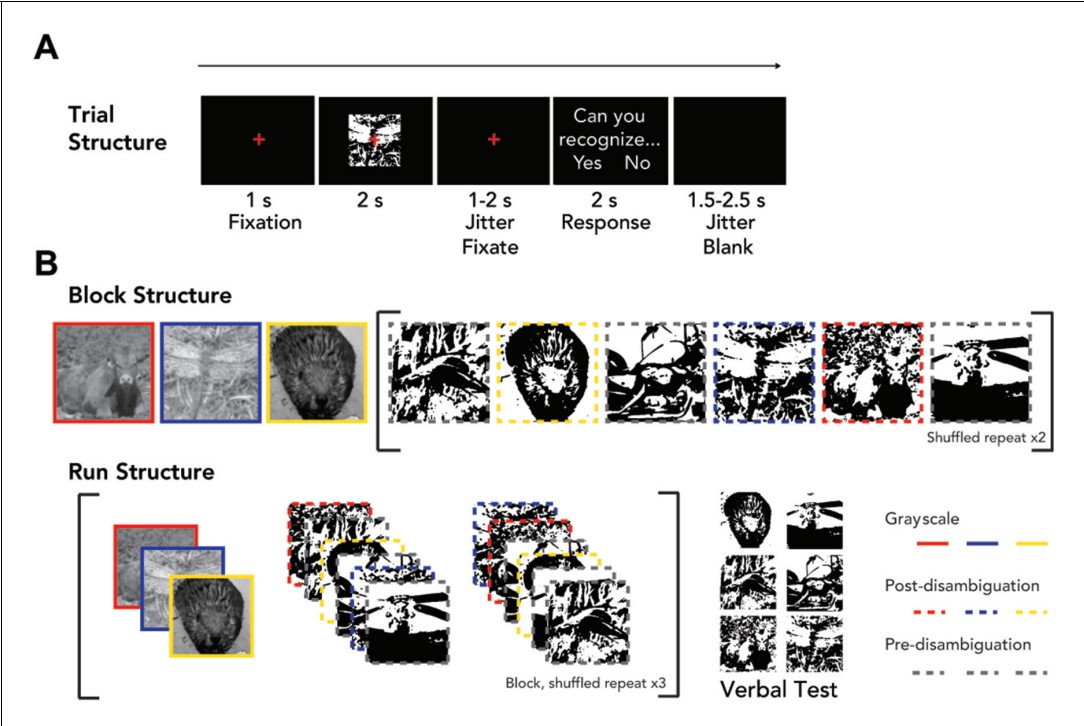

**Figure 1.** Task paradigm. (A) Trial structure. The left/right position (and corresponding button) for 'Yes'/'No' answer was randomized from trial to trial. (B) Block and run structure. Each block includes 15 trials: three grayscale images, six Mooney images in a randomized order, then a repeat of these six Mooney images in a randomized order. Three of the six Mooney images correspond to the grayscale images presented in the same run and are presented post-disambiguation. The other three Mooney images are presented pre-disambiguation, and their matching grayscale images will be shown in the following run. An experimental run consists of a block presented three times with randomized image order, followed by a verbal test (for details, see Materials and methods). Mooney images were not presented to subjects with colored frames.
DOI: https://doi.org/10.7554/eLife.41861.002

rates (pooled across six presentations) and correct verbal identification rates as the dependent variable (*Figure 2A*). There was a significant main effect of presentation stage (pre- vs. post-disambiguation) on both subjective recognition (*Figure 2A*, left, $F_{1,68}$ = 42.6, p=1.0 × $10^{-8}$, $\eta^2_p$ = 0.38) and verbal identification (*Figure 2A*, right, $F_{1,68}$ = 60.7, p=5.2 × $10^{-11}$, $\eta^2_p$ = 0.47). Crucially, there was also a significant interaction effect of presentation stage × image set type on both subjective recognition ($F_{1,68}$ = 16.4, p=1.0 × $10^{-4}$, $\eta^2_p$ = 0.19) and verbal identification ($F_{1,68}$ = 38.5, p=3.7 × $10^{-8}$, $\eta^2_p$ = 0.36), suggesting that the effect of disambiguation by viewing the corresponding grayscale image significantly exceeds that induced by repetition.

Second, given that each Mooney image was presented six times before and six times after disambiguation with subjective recognition probed each time, we assessed the effect of repetition on Mooney image recognition. To this end, we conducted a three-way repeated-measures ANOVA [factors: presentation number (1 – 6), presentation stage (pre- vs. post-disambiguation), and image-set type (real vs. catch)] on subjective recognition rates (*Figure 2B*). No main or interaction effect involving presentation number was significant (all p>0.2). However, consistent with the previous analysis, there was a highly significant interaction effect of presentation stage ×image set type ($F_{1,408}$ = 85.2, p=1.5 × $10^{-18}$, $\eta^2_p$ = 0.17). These results suggest that viewing the corresponding grayscale image, but not repeated viewing of the Mooney image, significantly facilitates Mooney image recognition.

Last, we examined the distribution of subjective recognition rates across individual Mooney images and subjects. The distribution of subjective recognition rates was separately plotted for pre- (*Figure 2C*, left) and post- (*Figure 2C*, right) disambiguation images, and for real (colored bars) and catch (empty bars) image sets. A bimodal distribution is observed in both stages. Accordingly, we defined those images recognized two or fewer times as 'not-recognized', and those recognized four or more times as 'recognized' (rectangles in *Figure 2C*). Based on these criteria, for real image sets,

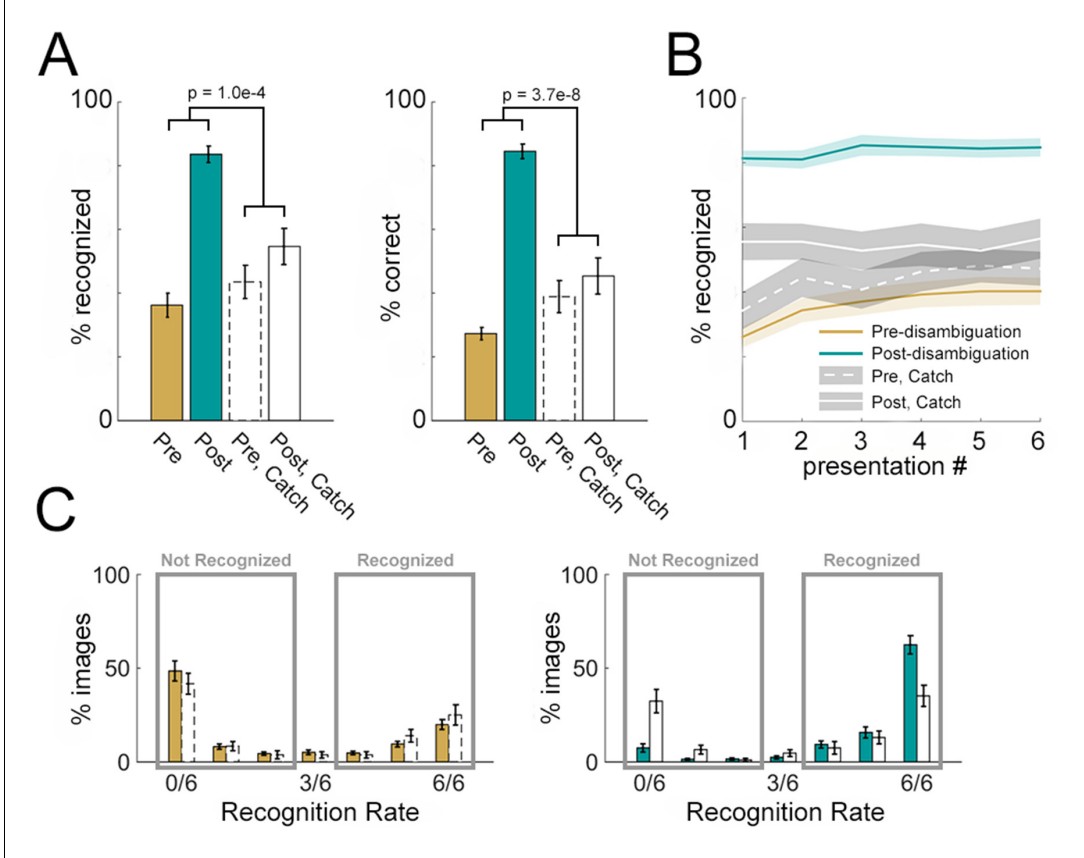

**Figure 2.** Disambiguation-related behavioral effects. (**A**) Subjective recognition rates (left, averaged across six presentations) and correct verbal identification rates (right) for Mooney images presented in the pre- and post- disambiguation period (gold and teal bars). Corresponding results for catch images are shown in open bars. p-Values corresponding to the interaction effect (pre vs. post × catch vs. non-catch) of two-way ANOVAs are shown in the graph. (**B**) Subjective recognition rates grouped by presentation number in the pre- and post- disambiguation stage (gold and teal), as well as the corresponding rates for catch images (white lines). (**C**) Distribution of subjective recognition rates across Mooney images in the pre- (gold bars, left) and post- (teal bars, right) disambiguation stage. Corresponding distributions for catch images are shown as open bars. All results show mean and s.e.m. across subjects.

DOI: https://doi.org/10.7554/eLife.41861.003

60.9 ± 4.2% are not recognized and 34.0 ± 3.4% are recognized in the pre-disambiguation stage, whereas only 10.3 ± 2.4% are not recognized and 87.4 ± 2.5% are recognized in the post-disambiguation stage.

Together, these results show the robustness of the Mooney image disambiguation effect, whereby having seen a related unambiguous image dramatically facilitates recognition for the degraded Mooney image. Importantly, in our paradigm a grayscale image is rarely followed by its corresponding Mooney image in the immediate next trial due to the block structure and the shuffling of image sequence (*Figure 1B*). This design and the long trial length (on average 8.5 s, see *Figure 1A*) ensured that the disambiguation effect cannot be driven by low-level priming (also see *Chang et al., 2016*).

## Disambiguation changes neural activity patterns elicited by Mooney images

To dissect the neural dynamics underlying the resolution of visual ambiguity by prior experience, we first investigated the neural activities that distinguish between pre- and post- disambiguation Mooney images regardless of image identity, using time-resolved multivariate decoding applied to whole-brain MEG data. This analysis was carried out using two frequency bands: slow cortical potentials (SCPs, 0.05–5 Hz) that were recently shown to be involved in conscious vision (*Li et al., 2014*;

*Baria et al., 2017*) and the classic event-related field (ERF, DC – 35 Hz) band that has been used to decode stimulus features under seen and unseen conditions (*Salti et al., 2015*; *King et al., 2016*). Using the 33 real image sets, we constructed three classifiers: First, all 33 unique Mooney images are included, and presentation stage (pre- vs. post-disambiguation) was decoded (*Figure 3A–B*, green). Second, because a small number of Mooney images (34.0 ± 3.4%) were spontaneously recognized pre-disambiguation, and 10.3 ± 2.4% of Mooney images remained unrecognized in the post-disambiguation period, we performed a behaviorally constrained analysis to more precisely target the disambiguation effect: for each subject, Mooney images that were both not-recognized (/not-identified) pre-disambiguation and recognized (/identified) post-disambiguation are selected. This resulted in 16.0 ± 1.4 (mean ±s.e.m. across subjects) real image sets based on subjective recognition (or 17.5 ± 1.0 real image sets based on verbal identification). Using these image sets, the presentation stage (pre- vs. post- disambiguation) was decoded (*Figure 3A–B*, blue and magenta). In a control analysis, the presentation stage of the catch image sets was also decoded, regardless of behavioral responses (*Figure 3A–B*, black traces).

With SCP activity (*Figure 3A*), the classifier using real image sets performed significantly above chance level at every time point from 300 ms onward (green trace and bar, p<0.05, cluster-based permutation test). Classifiers based on behavioral performance performed similarly (blue and magenta). By contrast, the classifier applied to catch image sets yielded chance-level accuracy throughout the trial epoch (black). A control analysis showed that decoding of presentation stage based on six randomly selected real image sets to match the statistical power of catch image sets still yielded sustained significant decoding from 300 ms onward (figure not shown). All the results are qualitatively similar but, as expected, with more high-frequency fluctuations, when the classifiers were constructed using the ERF band (*Figure 3B*).

In order to shed light on the neural activity contributing to successful decoding, we performed activation pattern transform on the classifier weights (*Haufe et al., 2014*). Activation patterns corresponding to the SCP and ERF classifiers based on subjective recognition responses (*Figure 3A–B*, blue traces) are plotted in *Figure 3C*, which reveal strong frontotemporal contributions starting within 400 ms after stimulus onset (note that both large positive and large negative values contribute strongly to the classifier). Later, we will probe the spatiotemporal evolution of relevant neural activities more precisely using model-driven MEG-fMRI data fusion.

## Cross-time generalization of decoding perceptual states

The above decoding analysis assesses whether there is separable neural activity pattern information between perceptual states at each time point following stimulus onset. To understand how this pattern information evolves over time, we next investigated decoder cross-time generalization (*King and Dehaene, 2014*). Here, a classifier trained at a given time point is tested on other time points in the trial epoch, and the extent to which it generalizes to another time point reveals how similar or different the underlying neural activity patterns are between the two time points. The temporal generalization matrix (TGM) corresponding to the subjective recognition classifier constructed with the SCP band (*Figure 3A*, blue trace) is shown in *Figure 3D*, left panel. Results obtained using the 'Pre vs. Post' classifier and the verbal identification classifier, or using the ERF band, were qualitatively similar (not shown). Significant generalization (p<0.05, cluster-based permutation test, outlined by the dashed trace) was obtained in a large temporal cluster that includes a wide diagonal beginning at ~300 ms, and a square shape from ~800 ms until the end of the trial epoch. The cross-time generalization accuracies of five classifiers trained at 400 ms intervals are shown as blue traces in *Figure 3D*, right panel, as compared with the within-time decoding accuracy, shown as black traces. For the classifier trained at 400 ms (bottom row), its cross-time generalization time course departs significantly from the within-time decoding time course (shown as shaded areas, p<0.05, paired t-tests, FDR corrected). By contrast, for the classifiers trained at 800 ms or later, their cross-time generalization accuracies closely tracked the within-time decoding accuracy. Together, these results suggest evolving patterns of neural activity until ~800 ms after stimulus onset and thereafter sustained activity patterns that distinguish pre- and post-disambiguation perceptual states.

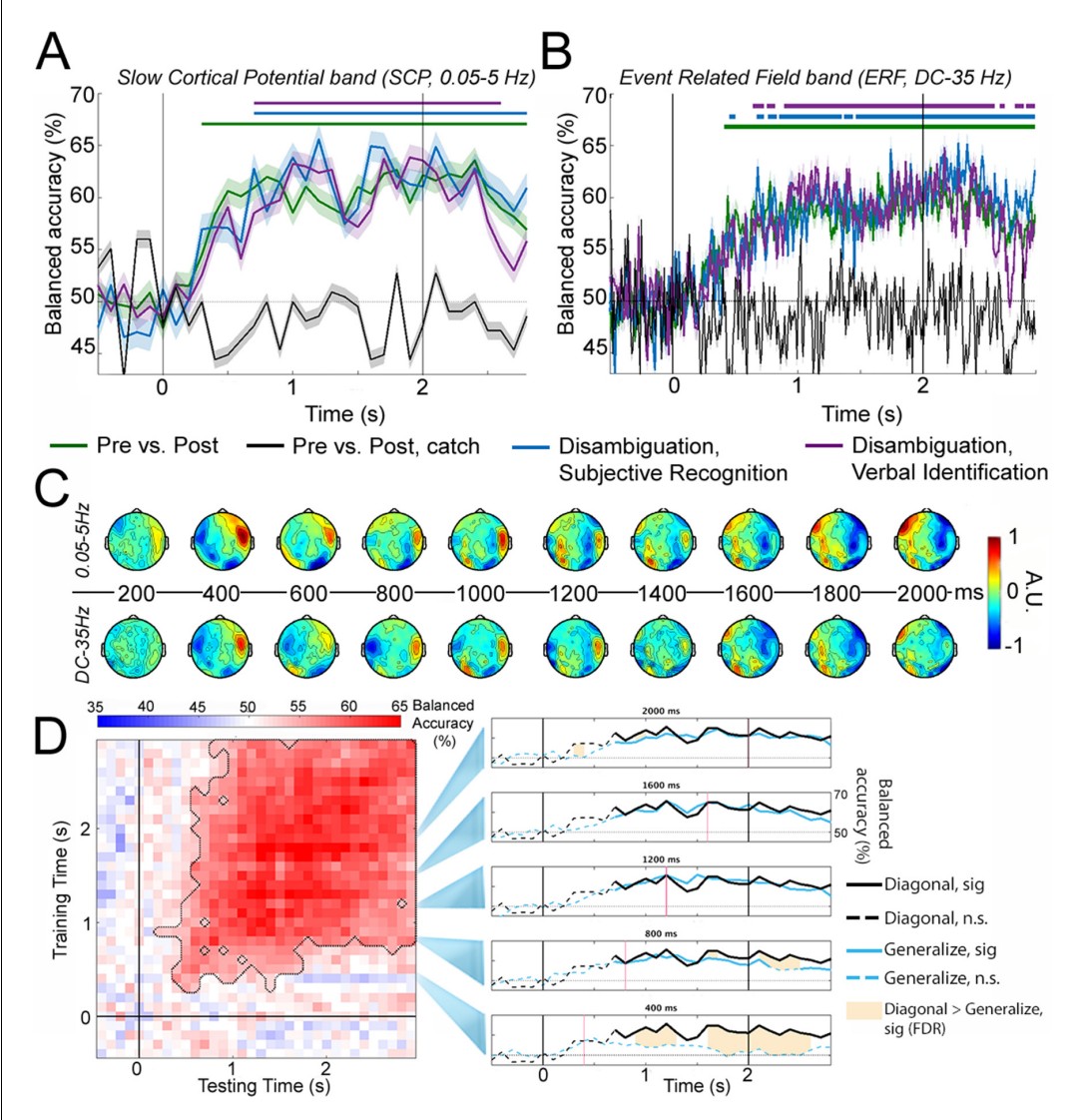

**Figure 3.** Decoding perceptual state information. (**A**) Decoding accuracy using SCP (0.05–5 Hz) activity across all sensors. Classifiers were constructed to decode i) presentation stage for all 33 Mooney images (Pre vs. Post; green); ii) presentation stage using only Mooney images that are not recognized pre-disambiguation and recognized post-disambiguation (Disambiguation, subjective recognition; blue); iii) presentation stage using only Mooney images that are not identified pre-disambiguation and identified post-disambiguation (Disambiguation, verbal identification; magenta); iv) 'presentation stage' for catch images, where the grayscale images did not match the Mooney images (black). Shaded areas represent s.e.m. across subjects. Horizontal bars indicate significant time points (p<0.05, cluster-based permutation tests). (**B**) Same as A, but for ERF (DC – 35 Hz) activity. (**C**) Activation patterns transformed from decoder weight vectors of the 'Disambiguation, subjective recognition' decoder constructed using SCP (top row) and ERF (bottom row) activity, respectively. (**D**) Left: TGM for the 'Disambiguation, subjective recognition' decoder constructed using SCP. Significance is outlined by the dotted black trace. Right: Cross-time generalization accuracy for classifiers trained at five different time points (marked as red vertical lines; blue traces, corresponding to rows in the TGM). The within-time decoding accuracy (corresponding to the diagonal of the TGM and blue trace in A) is plotted in black for comparison. Solid traces show significant decoding (p<0.05, cluster-based permutation test); shaded areas denote significant differences between within- and across- time decoding (p<0.05, FDR corrected). The black vertical bars in A, B, and D denote onset (0 s) and offset (2 s) of image presentation.

DOI: https://doi.org/10.7554/eLife.41861.004

## Disambiguation increases across-image dissimilarity in neural dynamics

The above results show separation in neural activity patterns between Mooney images presented in the pre- vs. post-disambiguation period, when the same physical images elicit distinct perceptual outcomes (meaningless blobs vs. recognizable animals or objects). However, these results do not

necessarily suggest image-content-specific processing: successful decoding of perceptual state may also reflect non-content-specific processing such as heightened attention and salience or reduced task difficulty associated with post-disambiguation images. To probe content-specific neural processing, we next turned to RSA (*Kriegeskorte et al., 2008a*) which allows for individual-image-level comparisons of neural activity patterns. At every time point in the trial epoch, we constructed a representational dissimilarity matrix (RDM), where each element contains the neural dissimilarity (quantified as 1 − Pearson's r, computed over all sensors, see Materials and methods, *RSA*) between two individual images (*Figure 4A*). Neural dissimilarity was calculated both for different images presented in the same condition (e.g., the Pre-Pre square of the RDM), and for (same or different) images presented in different conditions (e.g., the Pre-Post square of the RDM).

Group-average RDMs at five different time points are shown in *Figure 4B*. Two patterns can be seen: First, at 300–600 ms following stimulus input, dissimilarity across all image pairs is relatively low, presumably driven by visual-evoked activity that has a similar gross spatial pattern for all images. Second, after 600 ms, dissimilarity between individual Mooney images in the pre-disambiguation period (Pre-Pre square of the RDM) is visibly lower than between post-disambiguation Mooney images (Post-Post square) or between grayscale images (Gray-Gray square). We quantified these effects by averaging across elements within the upper triangles of the Pre-Pre, Post-Post and Gray-Gray squares of the RDM (*Figure 4C–i*) and plotting the mean across-image dissimilarity for each perceptual condition across time (*Figure 4D*). In all perceptual conditions, neural dissimilarity between individual images decreases sharply following stimulus onset and reaches the trough around 400 ms. However, from ~500 ms onward, across-image dissimilarity was substantially lower for pre-disambiguation Mooney images (gold) than for post-disambiguation Mooney images (teal) or grayscale images (black), and this difference is significant from ~1 s to the end of the trial epoch ($p < 0.05$, cluster-based permutation tests). The difference between post-disambiguation Mooney images and gray-images was not significant at any time point. A similar analysis applied to catch image sets did not yield any significant difference between pre- and post-disambiguation stages, or between Mooney and grayscale images (*Figure 4—figure supplement 1*).

To ensure that the increase of across-image dissimilarity following disambiguation was not driven by increased noise in the data, we re-conducted the analysis shown in *Figure 4D* using cross-validated Euclidean distance, a distance metric that is unaffected by changing levels of noise and only captures the signal component that is shared between different partitions of the data set (see Materials and methods, *Euclidean Distance*). The results, shown in *Figure 4—figure supplement 2*, reproduce the findings in *Figure 4D* and confirm that the increased across-image dissimilarity following disambiguation was not driven by a changing level of noise between perceptual conditions.

Together, these results suggest that, even though Mooney images presented in the two task stages are physically identical, following the acquisition of perceptual priors, different Mooney images are represented much more distinctly in neural dynamics, in fact as distinctly as the grayscale images.

This dramatic effect raises two important questions: 1) Is this effect driven by the neural representations of Mooney images shifting towards those of their respective grayscale counterparts? 2) Does this effect reflect image-content-specific processing at the single-trial level? To answer these questions, we next probe how neural representation for a particular Mooney image changes following disambiguation at the single-trial level.

## Comparing image-specific dynamic neural representations across perceptual conditions at the single-trial level

Here, we assess how disambiguation changes dynamical neural representations by analyzing single-trial separability of neural activity patterns elicited by the same (or matching Mooney-grayscale) image presented in different conditions. To this end, we used a measure akin to single-trial decoding to quantify separability (at the single-trial level) of neural activities elicited by two images (e.g. Images A and B in the same condition or different conditions, or Image A in different conditions). This measure calculates how much neural similarity across multiple presentations of the same image exceeds neural similarity between the two different images (i.e. $r_{within} − r_{between}$, *Figure 5A–i*, for details see Materials and methods, *Single-trial separability*). Unlike the 1 − r distance measure used in the previous analysis (*Figure 4*), this metric takes into account the within-image, across-trial

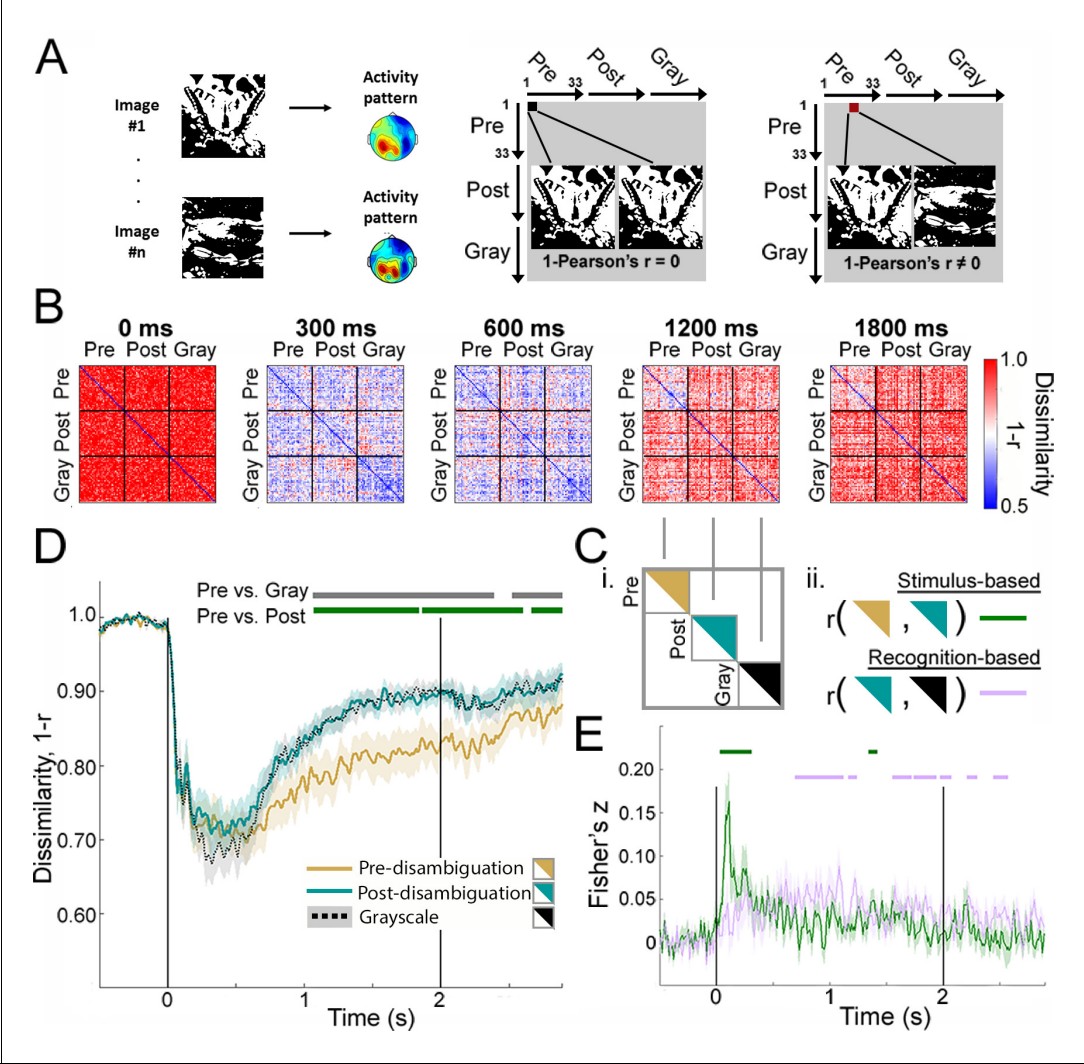

**Figure 4.** Image-level representations are influenced by prior information. (A) Schematic for RSA. See Materials and methods, *RSA*, for details. (B) Group-average MEG representation dissimilarity matrices (RDMs) at selected time points. (C) Schematics for analyses shown in D and E. (D) Mean across-image representational dissimilarity in each perceptual condition, calculated by averaging the elements in the upper triangles for each condition in the RDM (see C-i). Horizontal bars denote significant differences between conditions (p<0.05, cluster-based permutation tests). (E) Results from the intra-RDM analysis showing time courses of neural activity related to 'stimulus-based' and 'recognition-based' representation, obtained by performing element-wise correlations between the Pre and Post triangles in the RDM, and between the Post and Gray triangles, respectively (see C-ii). Correlation values were Fisher-z-transformed. Horizontal bars denote significance for each time course (p<0.05, cluster-based permutation tests). Shaded areas in D and E show s.e.m. across subjects.

DOI: https://doi.org/10.7554/eLife.41861.005

The following figure supplements are available for figure 4:

**Figure supplement 1.** Mean across-image representational dissimilarity in each perceptual condition for catch image sets.

DOI: https://doi.org/10.7554/eLife.41861.006

**Figure supplement 2.** Mean across-image representational dissimilarity in each perceptual condition for real image sets, calculated using Euclidean distance.

DOI: https://doi.org/10.7554/eLife.41861.007

variability: for instance, in the two examples given in *Figure 5A–ii* and $1 - r$ distance is identical, but single-trial separability ($r_{within} - r_{between}$) is higher in the top example.

We reconstructed the RDMs using the separability metric calculated across all MEG sensors. As shown in *Figure 5B*, there are darker diagonals in the between-condition squares: for example, in the Pre-Post square at 100 ms, and in the Post-Gray square at 800 ms. This suggests that single-trial

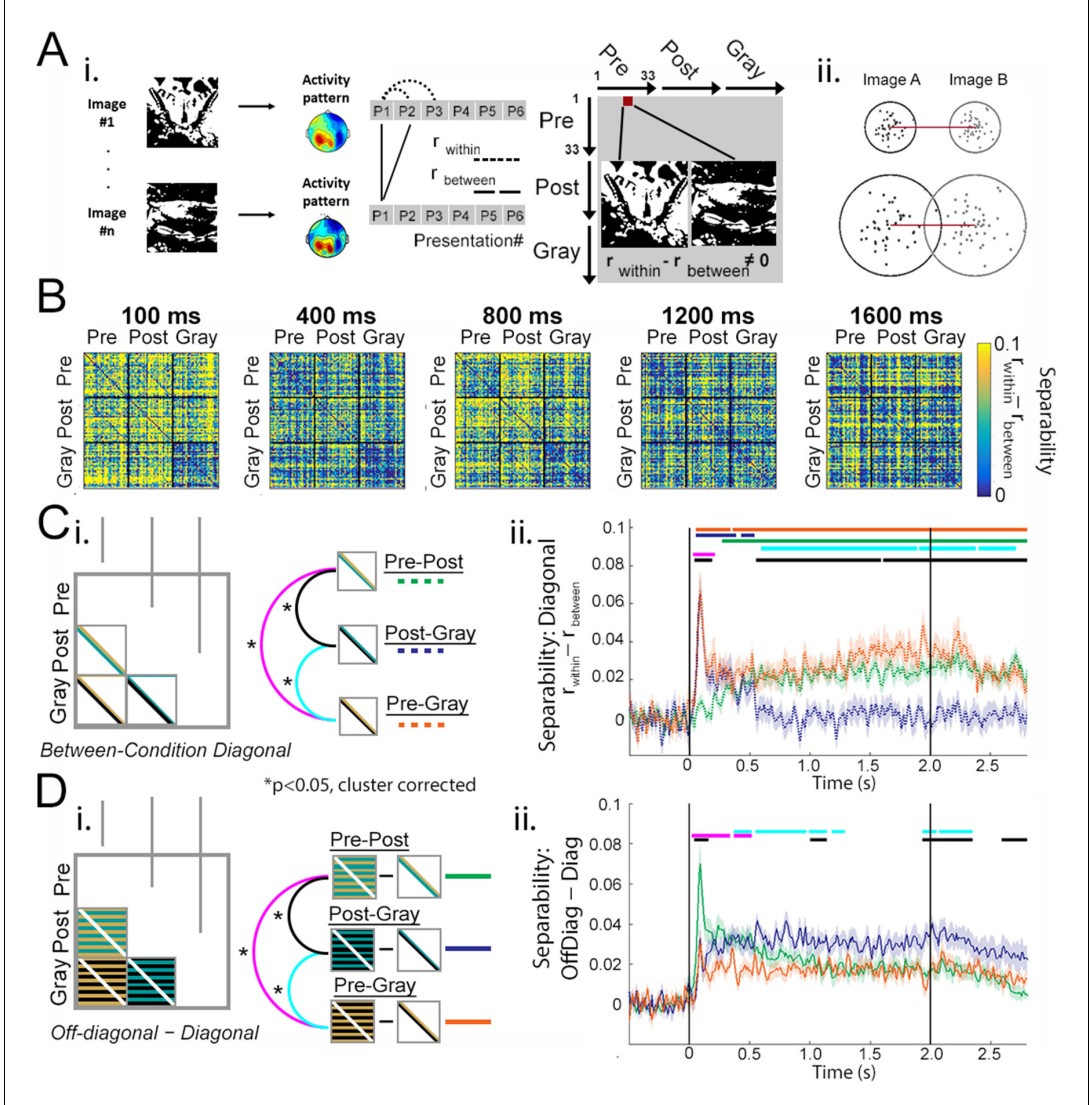

**Figure 5.** Single-trial separability analysis. (A) i: Schematic for separability calculation. For details, see Materials and methods, *Single-trial separability*. ii: Two hypothetical examples of single-trial neural activity patterns (projected to a 2-D plane) for Image A (black dots) and Image B (gray dots). The neural dissimilarity calculated based on trial-averaged activity patterns (1 – r measure used earlier) is identical between the two examples, while single-trial separability ($r_{within}$ - $r_{between}$) is higher in the top example. (B) Group-average MEG RDMs computed with the separability ($r_{within}$ - $r_{between}$) measure at selected time points. (C) Quantifying separability between neural activity patterns elicited by the same/matching image presented in different conditions. (i) Analysis schematic: diagonal elements in the between-condition squares of the RDM are averaged together, yielding three time-dependent outputs corresponding to the three condition-pairs. (ii) Separability time courses averaged across 33 real image sets for each between-condition comparison, following the color legend shown in C-i. The top three horizontal bars represent significant ($p<0.05$, cluster-based permutation test) time points of each time course compared to chance level (0); and the bottom three bars represent significance of pairwise comparisons between the time courses. (D) Quantifying the difference between off-diagonal and diagonal elements in the between-condition squares of the separability RDMs. Intuitively, this analysis captures how similar an image is to itself or its matching version presented in a different condition over and above its similarity to other images presented in that same condition. Statistical significance ($p<0.05$, cluster-based permutation test) for pairwise comparisons are shown as horizontal bars. When compared to chance (0), the three traces are significant from 40 ms (Pre-Post), 50 ms (Pre-Gray), and 60 ms (Post-Gray) onward until after image offset, respectively. Traces in C-ii- and D-ii show mean and s.e.m. across subjects.
DOI: https://doi.org/10.7554/eLife.41861.008

separability is lower between the same/matching image presented across conditions than between different images presented across conditions. Below we evaluate how disambiguation alters neural representation of the same image using two quantitative analyses applied to these RDMs.

First, we extracted the mean of diagonal elements within each between-condition square of the RDM (*Figure 5C–i*), which quantifies how separable neural activity patterns are between the same Mooney image presented before and after disambiguation (Pre-Post), between a pre-disambiguation Mooney image and its matching grayscale image (Pre-Gray), and between a post-disambiguation Mooney image and its matching grayscale image (Post-Gray). The result, plotted in *Figure 5C–ii*, shows that pre-disambiguation Mooney images are well separable from their matching grayscale images from 50 ms until the end of trial epoch (orange trace and significance bar, p<0.05, cluster-based permutation test). By contrast, post-disambiguation Mooney images are separable from their matching grayscale images only in a short window – from 50 to 530 ms (*Figure 5C–ii*, dark blue). Strikingly, neural representation of a post-disambiguation image is entirely indistinguishable from that of its matching grayscale image from ~550 ms onward (as shown by separability fluctuating around chance level; *Figure 5C–ii*, dark blue), despite the difference in the stimulus input. The separability between the same Mooney image presented before and after disambiguation starts to increase after stimulus onset, and reaches significance at 270 ms, which remains significant until the end of trial epoch (*Figure 5C–ii*, green). These results suggest that early (<300 ms) neural activity patterns distinguish between different physical stimulus inputs – Mooney vs. grayscale images, while late (>500 ms) neural activity patterns distinguish between recognizing the image content and failing to do so.

Second, we quantified the strength of between-condition diagonals (i.e. separability between the same/matching image presented across conditions) as compared to off-diagonal elements in the same between-condition squares (i.e. separability between different images presented across conditions), by computing the mean for each and calculating the difference between them (*Figure 5D–i*). This metric captures how similar an image is represented to itself or its matching version presented in a different condition over and above its similarity to other images presented in that same condition. The comparison between the Post-Gray traces in *Figure 5C–ii* and *Figure 5D–ii* (dark blue) reveals a striking effect after 550 ms: although neural activity patterns elicited by a post-disambiguation image are indistinguishable from those elicited by its matching grayscale image, they are well separable from other grayscale images, suggesting an image-specific shift in neural representation toward the relevant prior experience that guides perception. The difference between off-diagonal and diagonal separability is larger for the Pre-Post square than the Pre-Gray square from 20 to 520 ms (*Figure 5D–ii*, magenta), and larger for the Post-Gray square than the Pre-Gray square from 380 ms to 1.3 s (*Figure 5D–ii*, cyan).

Together, these results provide strong evidence that early (<300 ms) neural dynamics carry stimulus-content-specific processing and late (>500 ms) neural dynamics carry recognition-content-specific processing.

## Temporal separation of stimulus- and recognition-related neural representations

To further shed light on neural mechanisms underlying different information processing stages involved in prior-guided visual disambiguation, we investigated how the representational geometry (i.e. the set of representational distances between image-pairs) compares between perceptual conditions (for details see *Materials and methods*, *Intra-RDM analysis*). We reasoned that because the same set of Mooney images are presented in the Pre and Post stages, and they are ordered in the same sequence within the Pre-Pre and Post-Post squares of the RDM, neural activity reflecting stimulus-feature processing should exhibit a similar pattern between Pre-Pre and Post-Post squares of the RDM, and this can be quantified by performing an element-by-element correlation between these two portions of the RDM (*Figure 4C–ii*, 'stimulus-based' representation). Likewise, because the recognition content of a post-disambiguation Mooney image is similar to that of its corresponding grayscale image despite different stimulus input (e.g. 'it's a crab!'), neural activity reflecting recognition-content processing should exhibit a similar pattern between the Post-Post and Gray-Gray squares of the RDM, and this effect can be quantified by an element-by-element correlation between these two portions of the RDM (*Figure 4C–ii*, 'recognition-based' representation). For simplicity and ease of interpretation, for this analysis we used RDMs calculated based on trial-averaged activity patterns using the 1 − r measure (*Figure 4B*).

Neural activity reflecting stimulus processing (indexed by r[Pre-Pre,Post-Post]) exhibited an early sharp peak reaching significance at 30–310 ms following stimulus onset and a small second peak

that reaches significance at 1340–1420 ms (p<0.05, cluster-based permutation test; *Figure 4E*, green). By contrast, neural activity reflecting recognition processing (indexed by r[Post-Post,Gray-Gray]) occurs in a broad temporal period: it onsets shortly after stimulus onset and reaches significance at 690–1240 ms and again from 1550 ms to after stimulus offset (*Figure 4E*, magenta). As a control measure, the correlation between Pre-Pre and Gray-Gray squares of the RDM was not significant at any time point, suggesting that, as expected, representational geometry is different between conditions with different stimulus input and different recognition outcomes. In addition, an analysis using RDMs constructed with cross-validated Euclidean distances (for details, see Materials and methods, *Euclidean distance*) yielded similar results (figure not shown), suggesting that these findings are not driven by changing levels of noise between conditions. Nonetheless, likely due to insufficient statistical power, a direct contrast between r[Pre-Pre,Gray-Gray] and r[Post-Post,Gray-Gray] did not yield any significant time point following cluster-based correction. Thus, these results provide qualitative evidence – at the level of representational geometry – in accordance with our earlier conclusion that prior-guided visual disambiguation involves a dynamic two-part process including early stimulus-feature-related processing and late recognition-content-related processing. In the final analysis presented below, we will quantitatively test this possibility using a model-driven MEG-fMRI fusion analysis that simultaneously elucidates the spatial dimension of the evolving neural dynamics.

## Model-driven MEG-fMRI data fusion spatiotemporally resolves neural dynamics related to stimulus, attention, and recognition processing

In order to spatiotemporally resolve neural dynamics underlying different information processing stages in prior-guided visual perception, we applied a recently developed model-driven MEG-fMRI data fusion approach (*Hebart et al., 2018*). Nineteen additional subjects performed a similar Mooney image task, with an identical set of Mooney and grayscale images, during whole-brain 7T fMRI scanning (for details see Materials and methods). Our earlier MVPA and RSA results obtained from this fMRI data set suggested the involvement of frontoparietal (FPN) and default-mode (DMN) network regions in Mooney image disambiguation, in addition to early and category-selective visual areas (*González-García et al., 2018*). Based on these findings, 20 regions-of-interest (ROIs) were defined, covering early visual cortex (V1-V4), lateral occipital complex (LOC), fusiform gyrus (FG), and regions in the FPN and DMN (*Figure 6B*).

We designed three model RDMs to capture different information processing stages (*Figure 6A*):

*First*, a 'stimulus' model, which captures dissimilarity structure based on physical image features. This model includes three levels of dissimilarity: low (blue diagonal in the Pre-Post square, capturing dissimilarity between the same Mooney image presented pre- and post-disambiguation); medium (white off-diagonal elements in the Pre-Pre, Pre-Post, Post-Post, and Gray-Gray squares, capturing dissimilarity between different Mooney images and between different grayscale images); and high (red off-diagonal elements in the Pre-Gray and Post-Gray squares, capturing dissimilarity between Mooney and non-matching grayscale images). Thus, this model considers both gross image statistics (black-and-white Mooney vs. grayscale) and features specific to each image (the same Mooney image presented across conditions). For simplicity, the diagonals in the Pre-Gray and Post-Gray squares are excluded from this model, since we do not have an *a priori* judgment about these values in comparison to the others.

*Second*, a 'recognition' model, which aims to capture content-specific recognition processing. This model includes three levels of dissimilarity: low (blue off-diagonal elements in the Pre-Pre square and blue diagonal in the Post-Gray square; we note that the equivalence of these two categories was arbitrary); medium (white Pre-Post and Pre-Gray squares); and high (red off-diagonal elements in the Post-Post, Post-Gray and Gray-Gray squares). This model capitalizes on the intuition that the content of recognition is image-specific, and postulates that neural representations of two recognized images that have different contents are most distinct from each other, those of two unrecognized images are most similar to each other, and the dissimilarity is intermediate between a recognized and an unrecognized image. Moreover, since a post-disambiguation Mooney image yields a similar recognition content as its matching grayscale image, the model assumes a low dissimilarity between them.

*Third*, an 'attention' model, which captures dissimilarity structure based on the recognition *status*. This model includes two levels of dissimilarity: low (blue elements in the Pre-Pre, Post-Post, Gray-

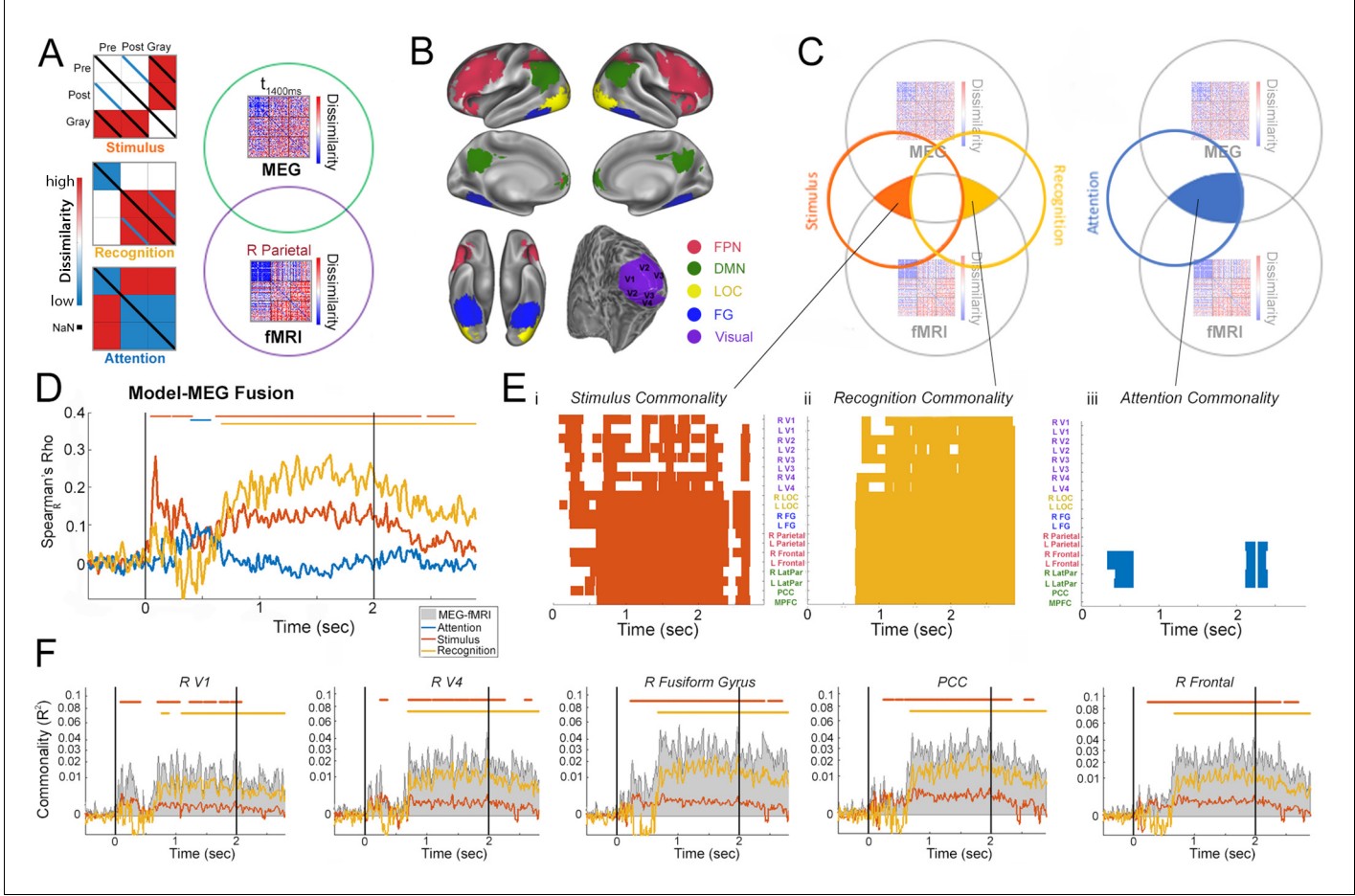

**Figure 6.** Model-based MEG-fMRI fusion analysis. (**A**) Model RDMs and analysis cartoon. *Left*: RDMs corresponding to 'Stimulus', 'Recognition', and 'Attention' models. For details, see *Results*. *Right*: MEG RDM from each time point and fMRI RDM from each ROI are compared, and shared variance between them that is accounted for by each model is computed. (**B**) ROIs used in the fMRI analysis. For details, see Materials and methods. These were defined based on a previous study (*González-García et al., 2018*). (**C**) Schematics for the commonality analyses employed in the model-based MEG-fMRI data fusion (results shown in E-F). Because neural activities related to the Stimulus and Recognition model overlap in time (see D), to dissociate them, variance uniquely attributed to each model was calculated (left). Shared variance between MEG and fMRI RDMs accounted for by the attention model is also assessed (right). (**D**) Correlation between model RDMs and MEG RDMs at each time point. Horizontal bars denote significant correlation (p<0.05, cluster-based permutation tests). (**E**) Commonality analysis results for Stimulus (**i**), Recognition (**ii**), and Attention (**iii**) models. Colors denote significant (p<0.05, cluster-based permutation tests) presence of neural activity corresponding to the model in a given ROI and at a given time point (with 10 ms steps). (**F**) Commonality time courses for the Stimulus (red) and Recognition (yellow) model (analysis schematic shown in panel C, left) for five selected ROIs, showing shared MEG-fMRI variance explained by each model. Total shared variance between MEG and fMRI RDMs for each ROI is plotted as gray shaded area. PCC: posterior cingulate cortex; R Frontal: right frontal cortex in the FPN. Horizontal bars denote significant model-related commonality (p<0.05, cluster-based permutation tests).

DOI: https://doi.org/10.7554/eLife.41861.009

The following figure supplement is available for figure 6:

**Figure supplement 1.** Stimulus and Recognition commonality results for all remaining ROIs.
DOI: https://doi.org/10.7554/eLife.41861.010

Gray, and Post-Gray squares); and high (red elements in the Pre-Post and Pre-Gray squares). This model postulates that non-recognized images (Pre) are represented similarly to each other, and recognized images (Post and Gray) are represented similarly to each other, while recognized and non-recognized images are represented differently. Thus, the 'attention' model captures changes induced by the status of recognition regardless of content, such as heightened attention or arousal that accompanies recognition.

We first assessed the correspondence between each model and the MEG RDM (computed using the 1 − r measure, see *Figure 4B*) at each time point (i.e. model-based RSA; for example see *Harel et al., 2014*; *Wardle et al., 2016*; *Vida et al., 2017*). The results reveal distinct temporal waveforms for neural activity related to each model (*Figure 6D*): the stimulus model dominates in the early period before 500 ms and exhibits a second broad plateau between 600 ms and 2.5 s, reaching statistical significance at 50–410 and 620–2410 ms (p<0.05, cluster-based permutation test). By contrast, the recognition model dominates in the late period, from ~500 ms to 2.5 s, reaching significance from 670 ms until the end of trial epoch. Interestingly, the attention model shows a peak at the transition between stimulus and recognition models, around 500 ms after stimulus onset, and reaches significance at 400–570 ms. We note that the waveforms of neural activity related to stimulus and recognition models qualitatively agree with the 'stimulus-based' and 'recognition-based' neural activity identified in the earlier intra-RDM analysis (*Figure 4E*), even though the methods employed by these two analyses are distinct.

To elucidate the brain regions contributing to each process, we conducted a model-driven MEG-fMRI data fusion analysis, using a commonality analysis approach (*Seibold and McPhee, 1979*; *Hebart et al., 2018*). Because neural activity related to the stimulus and recognition models overlapped in time (*Figure 6D*), we first performed an analysis to decompose the amount of shared variance between fMRI RDM from a given ROI and MEG RDM at a given time point that is uniquely explained by each model, while excluding the other model's effect (see schematic in *Figure 6C*, left, and *eq. 5* in Materials and methods). The results of this analysis are shown in *Figure 6Ei-ii*. The stimulus-related effects (*Figure 6E–i*) had the earliest onset in the right V1 and V2 at 80 ms, followed by progressive recruitment of areas along the visual hierarchy. Stimulus-related effects in most higher order frontoparietal regions reached significance much later (after 400 ms), with the exception of bilateral frontal cortices (right: 230 ms; left: 240 ms) and the PCC (240 ms), where significant effects occurred nearly simultaneously with the last visual areas. Interestingly, after 600 ms, stimulus-related effects exhibit sustained significance across category-selective visual areas (LOC and FG) and frontoparietal regions (FPN and DMN), and, at the same time, recurrent transient significance in early visual areas (V1-V4). This pattern may be driven by a continued cross-talk between higher-order and lower-order regions related to stimulus-content processing while the image is present.

By contrast, the recognition-related effects (*Figure 6E–ii*) exhibited broad spatiotemporal significance from ~650 ms until the end of trial epoch, covering all ROIs investigated but with earlier onset and more sustained significance in higher-order regions (LOC, FG, FPN and DMN) than early visual areas. Stimulus and recognition-related commonality time courses (i.e. shared MEG-fMRI variance explained by each model) are plotted in *Figure 6F* for five selected ROIs across the cortical hierarchy (see *Figure 6—figure supplement 1* for results from remaining ROIs).

Because neural activity related to the attention model did not overlap in time with the other two models (*Figure 6D*), we performed a second analysis to quantify the amount of shared variance between fMRI and MEG RDMs that is explained by the attention model (*Figure 6C*, right, and *eq. 4* in Materials and methods). Attention-related effects were only found in bilateral frontal and parietal cortices, which reached significance first in frontal cortices at 340 ms (*Figure 6E–iii*).

Together, these findings reveal the spatiotemporal evolution of neural activities underlying different information processing stages of prior-guided visual ambiguity resolution, including stimulus-feature, attentional-state, and recognition-content-related processing. Importantly, they show that stimulus-feature processing progresses from lower to higher order brain regions in an early time window, while recognition-related processing proceeds from higher to lower order regions in a later time period. Moreover, attention-related processing mediated by frontoparietal regions occurs at an intermediate time latency (~500 ms) and may facilitate the transition from stimulus processing to successful recognition.

## Discussion

Despite the pervasive need to resolve stimulus ambiguity (caused by occlusion, clutter, shading, and inherent complexities of natural objects) in natural vision (*Olshausen and Field, 2005*) and the enormous power that prior knowledge acquired through past experiences wields in shaping perception (*Helmholtz, 1924*; *Albright, 2012*), the neural mechanisms underlying prior-guided visual recognition remain mysterious. Here, we exploited a dramatic visual phenomenon, where a single exposure

to a clear, unambiguous image greatly facilitates recognition of a related degraded image, to shed light on dynamical neural mechanisms that allow past experiences to guide recognition of impoverished sensory input. Below we summarize our findings and discuss their implications.

Using whole-head MEG data, we first characterized the temporal evolution of neural activity patterns differentiating between perceptual states: pre- vs. post-disambiguation Mooney images, where identical stimulus input begets distinct perceptual outcomes depending on whether a perceptual prior is available. We observed that large-scale neural activity patterns (in the <35 Hz range) recorded by MEG reliably distinguished perceptual states starting from 300 ms after stimulus onset, and that these activity patterns are sustained from 800 ms onward. Consistent with an earlier study (*Baria et al., 2017*), slow cortical potentials (<5 Hz) accounted for much of this effect. These multivariate findings complement previous univariate observations showing perceptual state-related changes in higher frequency power and connectivity patterns (*Rodriguez et al., 1999*; *Grützner et al., 2010*; *Minami et al., 2014*; *Moratti et al., 2014*). However, to understand how perceptual priors interact with stimulus input to resolve recognition, identifying perceptual state-related changes in neural activity is far from sufficient. Such changes may reflect disambiguation-induced changes in attention, salience or task difficulty that are unrelated to perceptual processing of individual image content. To elucidate how disambiguation influences dynamic neural representation of individual images, we calculated time-resolved RDMs, which quantify the dissimilarity (1 − r; *Figure 4*), distance (cross-validated Euclidean distance; *Figure 4—figure supplement 2*), and single-trial separability ($r_{within} - r_{between}$; *Figure 5*) between neural activity patterns elicited by every image pair (within and across perceptual conditions), and performed fine-grained analyses based on these RDMs.

We first observed that across-image dissimilarity for post-disambiguation Mooney images rises higher than for their pre-disambiguation counterparts starting from ~500 ms onward (*Figure 4D*), and, surprisingly, closely parallels that for the grayscale images despite enormous differences in image features and the strength of bottom-up input between them.

To disentangle neural activities underlying different information processing stages involved in prior-guided visual perception, we compared the neural representational geometry (*Kriegeskorte and Kievit, 2013*) between perceptual conditions (*Figure 4E*). This analysis capitalizes on the intuition that neural activity encoding physical stimulus features should have identical representation for a Mooney image presented pre- and post-disambiguation. We identified the temporal evolution of neural dynamics fulfilling this criterion, which exhibits a sharp peak within ~300 ms following stimulus onset, and, interestingly, a second small peak that onsets around 800 ms and reaches significance at 1.3 s. Speculatively, this second peak may reflect top-down feedback related to filling in of perceptual details that follows the initial object recognition (*Ahissar and Hochstein, 2004*; *Campana and Tallon-Baudry, 2013*). Likewise, neural activity encoding the content of recognized objects (e.g. a crab, a motorcycle) should have similar representation between a post-disambiguation Mooney image and its matching grayscale image. The time course of such neural activity slowly increases following stimulus onset and exceeds stimulus-feature-related processing starting from ~500 ms.

We further investigated separability – at the single-trial level – between neural activity patterns elicited by the same image (or matching Mooney and grayscale images) presented in different perceptual conditions. The results revealed that within 500 ms after stimulus onset, neural activity patterns elicited by the same Mooney image pre- and post-disambiguation are substantially more similar to each other (i.e. lower separability) than to other images (*Figure 5D*, green); additionally, the neural activity pattern elicited by a Mooney image is significantly separable from that elicited by its matching grayscale image (*Figure 5C*, orange and dark blue). By contrast, after ~500 ms post-stimulus-onset, at the single-trial level, the neural activity pattern elicited by a post-disambiguation Mooney image is indistinguishable from that elicited by its matching grayscale image (*Figure 5C*, dark blue), yet they are significantly separable from other grayscale images (*Figure 5D*, dark blue). In the same late time period, the neural activity pattern elicited by a pre-disambiguation Mooney image is well separable from that elicited by the same image presented after disambiguation or by its matching grayscale image (*Figure 5C*, orange and green). These results provide clear evidence for a strong and specific shift of neural representation of each image toward the particular relevant prior experience, which gradually builds up after stimulus onset and is full-blown from ~500 ms onward.

Thus, multiple analyses probing neural representation format using dissimilarity, distance, and single-trial separability are in accordance with each other: they reveal content-specific neural processing of stimulus input in an early (<300 ms) time period, which transitions into content-specific neural processing related to recognition outcome in a late (>500 ms) time period. We note that the latency related to recognition observed in the current study is substantially later than previously reported onset times of object category and face identity-related information (*Carlson et al., 2013*; *van de Nieuwenhuijzen et al., 2013*; *Kaiser et al., 2016*; *Vida et al., 2017*). Yet, the late recognition-related neural dynamics observed herein are consistent with human subjects' reaction times reporting recognition status for disambiguated Mooney images at around 1.2 s (*Hegdé and Kersten, 2010*). Several aspects likely contribute to this difference: first, bottom-up stimulus information is ambiguous and much weaker for Mooney images than images showing clear and isolated objects typically used in object categorization tasks; second, our analyses probe neural activity related to recognizing individual image content rather than object category, and processing of face identity likely benefits from a specialized circuitry; third, neural activity differentiating between object categories may also reflect early processing of low-level image features that differ between categories (*Coggan et al., 2016*). Importantly, the present finding of slow neural dynamics underlying recognition-related processing is consistent with our hypothesis that recruitment of perceptual templates encoded in higher order frontoparietal areas and long-distance recurrent activity are needed for prior-experience-guided visual ambiguity resolution. This hypothesis receives further support from the model-driven MEG-fMRI fusion analysis, which we discuss below.

We capitalized on the ability of RSA to project high-dimensional neural data from different modalities into a common representational space (*Kriegeskorte et al., 2008a*) to combine the high temporal resolution of MEG data with high spatial resolution of a separate 7T fMRI data set. In addition, we adopted a recently developed model-driven data fusion approach building on RSA (*Hebart et al., 2018*) to spatiotemporally resolve neural activity underlying different information processing stages. To this end, we constructed three model RDMs that capture the representation format of idealized processes related to stimulus feature, recognition content, and attentional state processing (*Figure 6A*). Although these models capture relatively coarse features in the data, the time courses of MEG activity related to each model (*Figure 6D*) are consistent with the earlier fine-grained analysis applied to MEG RDMs, including the dominance of stimulus and recognition processing in the early (<500 ms) and late (>500 ms) time period, respectively, and a late second peak in stimulus-related processing. Interestingly, neural activity related to the attention model, which captures dissimilarity driven by recognition status (recognized vs. non-recognized), shows a single peak around 500 ms, suggesting that a transient salience signal may facilitate recognition processing of individual image content.

The model-driven MEG-fMRI data fusion analysis dissects the shared variance between MEG RDM at a given time point and fMRI RDM from a given brain region that is uniquely accounted for by each model. Consistent with our earlier fMRI findings (*González-García et al., 2018*), recognition-related neural activity is widely distributed across the cortical hierarchy – from early visual areas to frontoparietal and default-mode networks (*Figure 6E–ii*). Importantly, this analysis reveals that recognition-related activity reaches significance at an earlier time in higher-order regions (LOC, FG, FPN and DMN, 670–680 ms) than in lower-order regions (V1-V3, 760 ms or later). This sequence is consistent with our hypothesis that higher-order brain regions are crucial for encoding perceptual priors and initiating prior-guided recognition.

We note that bilateral frontal cortices of the FPN are the only regions showing both attention-related activity and early stimulus-related activity that reaches significance at 230–240 ms – immediately following fusiform gyri at 220–230 ms. This result resonates with our previous fMRI observation that following disambiguation, frontal areas of the FPN move up the cortical hierarchy as defined by the neural representation format (*González-García et al., 2018*). Together, these findings support the idea that frontal areas may play a special role in utilizing internal priors to guide perceptual processing (*Bar et al., 2006*; *Summerfield et al., 2006*; *Wang et al., 2013*).

In conclusion, the present results reveal, for the first time, how neural activities underlying different information processing stages during prior-guided visual perception dynamically unfold across space and time. These findings significantly further our understanding of the neural mechanisms that endow previous experiences with enormous power to shape our daily perception. In line with theories positing aberrant interactions between internal priors and sensory input in psychiatric illnesses

(*Friston et al., 2014*), behavioral and neural abnormalities associated with Mooney image disambiguation have been reported in patients with schizophrenia and autism (*Sun et al., 2012*; *Rivolta et al., 2014*; *Teufel et al., 2015*). Thus, our findings may also inform studies on the pathophysiological processes involved in these debilitating illnesses.

## Materials and methods

### Subjects

The experiment was approved by the Institutional Review Board of the National Institute of Neurological Disorders and Stroke (under protocol #14 N-0002). All subjects were right-handed and neurologically healthy with normal or corrected-to-normal vision. Eighteen subjects between 21 and 33 years of age (mean age 26.2; nine females) participated in the MEG experiment. Nineteen additional subjects (age range = 19–32; mean age = 24.6; 11 females) participated in a 7T fMRI experiment, using a similar task paradigm as in the MEG experiment (*González-García et al., 2018*). The two subject groups did not have any overlap since subjects needed to be naïve to the Mooney images used in this experiment. All subjects provided written informed consent.

### Stimuli

Mooney and grayscale images were generated from grayscale photographs of real-world objects and animals selected from the Caltech ([http://www.vision.caltech.edu/Image_Datasets/Caltech101/Caltech101.html](http://www.vision.caltech.edu/Image_Datasets/Caltech101/Caltech101.html)) and Pascal VOC ([http://host.robots.ox.ac.uk/pascal/VOC/voc2012/index.html](http://host.robots.ox.ac.uk/pascal/VOC/voc2012/index.html)) databases. Grayscale images were constructed by cropping photographs of objects and animals in a natural setting to $500 \times 500$ pixels and applying a box filter. Mooney images were constructed by thresholding the grayscale images. Threshold level and filter size were initially set at the median intensity of each image and $10 \times 10$ pixels, respectively. Each parameter was titrated so that the Mooney image was difficult to recognize without first seeing the corresponding grayscale image. From an original set of 252 images, thirty-nine (19 were inanimate objects, and 20 were animals – unbeknownst to the subjects) were chosen to be used in this experiment via an initial screening procedure conducted by six additional participants recruited separately from the main experiment [for details, see (*Chang et al., 2016*). Stimuli were presented using E-Prime Software (Psychology Software Tools, Sharpsburg, PA) via a Panasonic PT-D3500U projector with a ET-DLE400 lens. All images subtended $11.9 \times 11.9$ degrees of visual angle.

### Task paradigm

Each trial began with a 1 s fixation period during which subjects fixated on a red cross in the center of the screen (*Figure 1A*). Thereafter, a Mooney image or a grayscale image was presented for 2 s. The red cross was present during image presentation, and subjects were instructed to keep their gaze fixated whenever it was onscreen. After image presentation, there was another fixation period, the duration of which was uniformly distributed between 1 and 2 s. This was followed by a response prompt of 'Can you recognize the object hidden in the image?' to assess subjective recognition of each image presentation. Below this prompt, the answer choices, 'Yes' and 'No' were presented on each side of the screen with their positions randomly varied across trials. Subjects were instructed to answer the question by pressing one of two buttons using their right thumb, with each button corresponding to one side of the screen. The response prompt terminated when a response was given, and each trial ended with a blank screen of jittered duration uniformly distributed between 1.5 and 2.5 s.

Trials were organized into blocks, using a structure similar to previous studies (*Gorlin et al., 2012*; *Chang et al., 2016*). Each block consisted of 15 trials: three different grayscale images followed by six Mooney images, then a repeat of the same six Mooney images (*Figure 1B*). Three of the Mooney images corresponded to the preceding grayscale images ('post-disambiguation') and the other three were novel ('pre-disambiguation'). The presentation order of these six Mooney images in each repeat was randomized. The block was repeated twice with shuffled grayscale image sequences and shuffled Mooney image sequences, followed by a verbal test session. This constituted one experimental run. Grayscale images corresponding to pre-disambiguation Mooney images were presented in the subsequent run. In total, each participant completed 14 runs. For all Mooney

images to be presented both pre- and post- disambiguation, the first and last runs were half runs. The first run included only three pre-disambiguation Mooney images. The final run included three post-disambiguation Mooney images and their grayscale counterparts. All other full runs consisted of three grayscale images, three post-disambiguation Mooney images, and three novel pre-disambiguation Mooney images. In all, each unique grayscale image was presented three times for each subject, and each unique Mooney image was presented six times before and six times after disambiguation. The full experiment lasted approximately 2 hr.

The verbal test was included to verify that subjects' recognition of Mooney images was the correct interpretation. It consisted of presenting the six different Mooney images from the preceding run for 2 s each on the screen and participants were asked to verbally respond what they saw in the image. They were allowed to answer 'I don't know'. Verbal responses were scored as correct or incorrect using a pre-determined list of acceptable answers for each image. Subjects were verbally tested on each Mooney image once before disambiguation and once after disambiguation. No MEG signal was recorded during the verbal test.

Of the 39 unique Mooney images used for the main experiment, 33 had their corresponding grayscale images presented ('real image sets'), while the remaining six were presented with non-matching grayscale images ('catch image sets') as controls. The same set of real and catch images were used for all subjects. Details on statistical analyses can be found in the following sections and in the Results.

## MEG data acquisition

While performing the task, subjects' brain activity was recorded with a 275-channel whole-head MEG system (CTF), and their gaze position and pupil size were recorded using a SR Research Eyelink 1000+ system. Eye-tracking was used for online monitoring of fixation during the experiment. Three dysfunctional MEG sensors were removed from all analyses. MEG data were collected with a 600 Hz sampling rate and an anti-aliasing filter of 150 Hz. Before and after each run, the head position of a subject was measured using fiducial coils. Subjects responded to subjective recognition questions using a fibreoptic response button box. All MEG data samples were corrected with respect to the presentation delay of the projector (measured with a photodiode).

## MEG data preprocessing

The FieldTrip package implemented in MATLAB (*Oostenveld et al., 2011*) was used for preprocessing in conjunction with custom-written code in MATLAB (Mathworks, Natick, MA). MEG data were demeaned, detrended, and filtered in two different frequency bands (using $3^{rd}$-order Butterworth filters) for further analyses: slow cortical potentials (SCPs, 0.05 – 5 Hz; down-sampled to 10 Hz), and classic event-related field (ERF) frequency range (DC – 35 Hz; down-sampled to 100 Hz). Independent component analysis (ICA) was applied to continuous data from each run, and components corresponding to eye blinks, eye movements, and heartbeat-related artifacts were removed. Data were then epoched into 3.5 s trials consisting of a 0.5 s pre-stimulus period and a 3 s post-stimulus period (including 2 s image presentation and the first sec of jitter-fixate period; see *Figure 1A*). Baseline correction was applied for each sensor using the pre-stimulus time window.

## Multivariate pattern analysis (MVPA)

MVPA was carried out using both the 0.05 – 5 Hz data and the DC – 35 Hz data. First, MEG activity at each time sample was normalized across sensors (*Pereira et al., 2009*). For each subject, classification of perceptual state (pre- vs. post-disambiguation) was performed using activity from all MEG sensors averaged across six presentations for each image in each perceptual state, using all 33 non-catch image sets. We implemented a linear support vector machine (SVM) classifier (cost = 1) using the LIBSVM package (*Chang and Lin, 2011*) at each time point in the trial epoch. An odd-even cross-validation scheme was used, classification accuracy was averaged across the two folds and reported as balanced accuracy (*Brodersen et al., 2013*). Activation patterns corresponding to the MEG activity contributing to the classifier were computed for each subject and time point by multiplying the vector of SVM decoder weights with the covariance matrix of the data set used to train a given classifier (*Haufe et al., 2014*). For display purposes, activation patterns were averaged across subjects. To test the cross-time generalization of classifiers, a classifier trained at a given time point

was tested at all time points in the trial epoch, yielding a temporal generalization matrix (TGM) (*Stokes et al., 2013*; *King and Dehaene, 2014*). If a classifier can generalize across time points, this demonstrates that the decoded information format is similar across these time points. If it does not generalize, this indicates that the information is represented differently or not at all. A control analysis was carried out in a similar manner but using only the six catch image sets, for which pre- and post- states were defined as before and after the presentation of the artificially-assigned, non-matching grayscale images. An additional analysis (disambiguation decoding) used only non-catch image sets where Mooney image were unrecognized/non identified in the pre-disambiguation stage and recognized/identified in the post-disambiguation stage. These image sets were selected for each individual subject based on the behavioral responses.

## Cluster-based permutation tests for multivariate pattern decoding

The group-level statistical significance of classifier accuracy at each time point was assessed by a one-tailed, one-sample Wilcoxon signed rank test against chance level (50%). To correct for multiple comparisons, we used cluster-based permutation tests (*Maris and Oostenveld, 2007*). Temporal clusters were defined as contiguous time points yielding significantly above-chance classification accuracy ($p<0.05$). The test statistic W of the Wilcoxon signed rank test was summed across time points in a cluster to yield a cluster's summary statistic. Cluster summary statistics were compared to a null distribution, constructed by shuffling class labels 500 times, and extracting the largest cluster summary statistic for each permutation. Clusters in original data with summary statistics exceeding the 95th percentile of null distribution were considered significant (corresponding to $p<0.05$, cluster-corrected). For classifier temporal generalization, the permutation-based approach for cluster-level statistical inference used the same procedure as above, where clusters were defined as contiguous time points in both training and generalization dimensions with significant ($p<0.05$) above-chance classification accuracy.

## Representational similarity analysis (RSA)

For this and following analyses, to achieve higher temporal resolution, we used data filtered in the DC – 35 Hz range (down-sampled to 100 Hz). Similar to the preprocessing for MVPA analysis, MEG data were normalized across sensors at each time point. For each subject, data were averaged across the three presentations for each grayscale image. In order to compare between Mooney images and grayscale images, for each Mooney image the first three presentations were averaged together in the pre- and post-disambiguation stage, respectively. At each time point in the trial epoch, we computed representational distance, calculated as 1 – Pearson's r (computed across all sensors), between every image pair in all presentation stages: pre-disambiguation Mooney, post-disambiguation Mooney, and grayscale (*Figure 4A*). This generated a 99 × 99 representational dissimilarity matrix (RDM) at each time point (with 10 ms steps). Catch image sets were excluded from this analysis.

Using the time-resolved RDM, we averaged across all image-pairs within each presentation condition (i.e. off-diagonal elements within the Pre-Pre, Post-Post, and Gray-Gray squares of the RDM, see *Figure 4C–i*) to yield a mean dissimilarity time course for each condition. Dissimilarity time courses were compared between conditions using a Wilcoxon signed-rank test across subjects and corrected for multiple comparisons using a cluster-based permutation test similar to that described above, with 5000 shuffles of class labels for each subject.

## Euclidean distance

To ensure that the results were unaffected by changing levels of noise between perceptual conditions, the above analysis was repeated using RDMs constructed with Euclidean distance and cross-validated Euclidean distance. Let x and y represent neural activity vectors elicited by two different images (or the same image presented in two different conditions), (non-cross-validated) Euclidean distance is calculated as

$$d^2_{Euclidean}(x,y) = (x-y)(x-y)^T, \tag{1}$$

and cross-validated Euclidean distance is calculated as

$$d^2_{Euclidean, \, c.v.}(x,y) = (x-y)_{[A]}(x-y)^T_{[B]},$$ (2)

where A and B denote the two partitions of the data within each cross-validation fold (*Guggenmos et al., 2018*). We used a three-fold cross-validation scheme to calculate cross-validated Euclidean distance. With cross-validation, the contribution by noise in the data cancels out, and the result is only driven by the component in the data that is consistent across partitions.

## Intra-RDM analysis

To probe fine-grained information available in the RDMs, we performed an intra-RDM analysis (for detailed rationale, see Results). This analysis contained two components: First, to assess neural activity related to processing the physical features of individual images, we calculated Pearson's correlation between the Fisher-z transformed values of Pre-Pre and Post-Post squares of the MEG representational similarity matrices (RSMs, equal to 1-RDM) at each time point, using the upper triangle of each (*Figure 4C–ii*, 'Stimulus-based'). Second, to assess neural activity related to the recognition outcomes of individual images, we calculated Pearson's correlation between the Fisher-z transformed values of Post-Post and Gray-Gray squares of the MEG RSM, again using the upper triangle of each (*Figure 4C–ii*, 'Recognition-based'). The correlation values were Fisher-z transformed, and group-level statistics were assessed by one-sample t-tests against zero followed by cluster-based permutation tests with 5000 permutations.

## Single-trial separability

To assess how well neural activities elicited by Image A and Image B (importantly, these can represent two different images presented in the same or different conditions, or the same/matching image presented in different conditions) can be separated at the single-trial level, we computed a separability measure as follows.

Let $x_i$ and $y_i$ be the activity vectors across all sensors on the $i$-th presentation of Image A and Image B, respectively. And suppose that Image A and Image B are presented for a total of $m$ and $n$ trials, respectively. (Each Mooney image is presented six times before and six times after disambiguation, and each grayscale image is presented three times total.) We calculated the following measures:

$$r_{within} = \frac{2}{m(m-1)}\sum_{i=1}^{m-1}\sum_{j=i+1}^{m}r_z(x_i,x_j) + \frac{2}{n(n-1)}\sum_{i=1}^{n-1}\sum_{j=i+1}^{n}r_z(y_i,y_j)$$

$$r_{between} = \frac{1}{mn}\sum_{i=1}^{m}\sum_{j=1}^{n}r_z(x_i,y_j)$$

where $r_z$ denotes Fisher-z-transform of Pearson's correlation r-value. Separability is calculated as

$$Separability_{A,B} = r_{within} - r_{between}$$ (3)

Thus, separability quantifies how much neural similarity across multiple presentations of the same image exceeds neural similarity between the two different images and is akin to single-trial decoding.

We constructed the time-resolved 99 × 99 RDMs using the separability measure, which were subjected to two further quantitative analyses. In the first analysis, we extracted the mean of diagonal elements in the between-condition squares, which yielded three time-dependent outputs (*Figure 5C–i*). In the second analysis, we computed the difference between the mean off-diagonal value and the mean diagonal value within each between-condition square, which again yielded three time-dependent outputs (*Figure 5D–i*). For each analysis, we evaluated the statistical significance of each output against chance level using a one-sample t-test against 0, and the statistical significance of pairwise comparisons between the outputs using Wilcoxon sign-rank tests; all statistical tests were corrected for multiple comparisons using cluster-based permutation tests with 5000 permutations.

## 7T fMRI data collection and regions of interest (ROI) definition

We carried out an fMRI study using a similar paradigm, which included the same 33 Mooney images and their grayscale counterparts that made up the non-catch image sets in the MEG experiment. Run and block structure were identical to the MEG experiment. In the fMRI experiment, each trial included a 2 s fixation period, a 4 s image presentation, a 2 s blank period, and a 2 s response period to assess subjective recognition (similar question as in the MEG experiment). Detailed methods and results related to the fMRI study are reported separately (*González-García et al., 2018*); here we describe data collection and ROI definition procedures relevant to the current study.

fMRI data were collected on a Siemens 7T scanner equipped with a 32-channel head coil (Nova Medical). T1-weighted anatomical images were obtained using an MP-RAGE sequence (sagittal orientation, $1 \times 1 \times 1$ mm resolution). Additionally, a proton-density (PD) sequence was used to obtain PD-weighted images also with $1 \times 1 \times 1$ mm resolution, to help correct for field inhomogeneity in the MP-RAGE images (*Van de Moortele et al., 2009*). Functional images were obtained using a single-shot echo planar imaging (EPI) sequence (TR = 2000 ms, TE = 25 ms, flip angle = 50°, 52 oblique slices, slice thickness = 2 mm, spacing = 0 mm, in-plane resolution = $1.8 \times 1.8$ mm, FOV = 192 mm, acceleration factor/GRAPPA = 3). The functional data were later resampled to 2 mm isotropic voxels. Respiration and cardiac data were collected using a breathing belt and a pulse oximeter, respectively, simultaneously with fMRI data acquisition; and physiological noise were removed during pre-processing of fMRI data using the RETROICOR method (*Glover et al., 2000*). Anatomical and functional data preprocessing followed standard procedures and are described in detail in *González-García et al. (2018)*.

ROIs were defined as follows. A separate retinotopic localizer and a lateral occipital complex (LOC) functional localizer were performed for each subject to define bilateral early visual ROIs (V1, V2, V3, and V4) and LOC, respectively. Fusiform gyrus (FG) ROIs were extracted using the Harvard-Oxford Cortical Structural Atlas (https://fsl.fmrib.ox.ac.uk/fsl/fslwiki/Atlases). Default mode network (DMN) regions, including bilateral lateral parietal cortices (LatPar), medial prefrontal cortex (mPFC), and posterior cingulate cortex (PCC), were defined using a general linear model (GLM) of the disambiguation contrast (pre-disambiguation-not-recognized vs. post-disambiguation-recognized). Lastly, statistical map from searchlight MVPA decoding of the disambiguation contrast was used to define the frontoparietal network (FPN) ROIs: bilateral frontal and parietal cortices. The localization of all ROIs are shown in *Figure 6B*; for further details see (*González-García et al., 2018*). Critically, none of the analyses used to define the ROIs depended on the RDMs used for MEG-fMRI data fusion analysis described below; in addition, results obtained using FPN and DMN ROIs defined based on an independent resting-state study (*Power et al., 2011*) were similar (not included due to length consideration).

For each of the 20 ROIs, a $99 \times 99$ RDM was constructed using the activation patterns corresponding to each image in each condition, using the same image order as for the MEG RDM. These activation patterns were derived from a GLM and averaged across image presentations (three presentations for grayscale images; only the first three presentations were used for Mooney images in each stage in order to equalize statistical power across conditions). Representational distance between every image pair was computed as 1 – Pearson's r (computed across all voxels within an ROI), similarly as for MEG RDM.

## Model-based representational similarity analysis (RSA) of MEG data

RSA enables neural representation format to be compared with theoretical models of information representation (*Kriegeskorte et al., 2008a*). Here, we used *a priori* defined models to probe how neural representation of different types of information dynamically evolved over time. Three models were defined as RDMs (same dimensions as the MEG RDMs) where high dissimilarity was expressed as 1 (*Figure 6A*, *red*) and low dissimilarity was expressed as a 0 (*Figure 6A*, *blue*), with intermediate dissimilarity expressed as 0.5 (*Figure 6A*, *white*). Black diagonals in these model RDMs (*Figure 6A*) denote elements excluded from the model's analysis (defined as NaN's).

For each model and time point, Spearman correlation was computed between the upper triangles of model RDM and group-averaged MEG RDM to assess how well the model explained the MEG RDM (*Figure 6D*). Statistical significance was established using a cluster-based permutation test. The null distribution was calculated using 1000 permutations of the MEG RDM, where for each

permutation the image order was shuffled, but with the same shuffled order along the x- and y- axis of the RDM and across all time points. Clusters were defined as contiguous time points where the Spearman rho value was greater than the 95[th] percentile of the null distribution. To identify significant clusters, we determined the 95[th] percentile of maximum cluster size across all permutations, and clusters in the original data that exceeded this cut-off were deemed significant (equivalent to p<0.05, one-sided).

## Model-based MEG-fMRI data fusion

Following previous studies (*Kriegeskorte et al., 2008b*; *Cichy et al., 2014*), we employed cross-modal RSA to combine fMRI and MEG data from independent participant groups (N = 19 and 18, respectively). Furthermore, we applied a recently developed approach based on commonality analysis (*Seibold and McPhee, 1979*; *Hebart et al., 2018*) to use the theoretical models described in the previous section to guide the cross-modal MEG-fMRI data fusion.

Specifically, commonality analysis was employed in two ways. First, it was used to determine the shared variance between a model (M1 in *eq. 4*), the fMRI RDM from a given ROI and the MEG RDM at a given time point (*Figure 6C*, right), calculated as:

$$C(MEG,fMRI,M1) = R^2_{MEG,fMRI} + R^2_{MEG,M1} - R^2_{MEG,fMRI,M1} \tag{4}$$

Second, it was used to determine the shared variance between a model, the fMRI RDM and the MEG RDM which is unique to that model, and is not shared by a second model (*Figure 6C*, left). This allows the dissociation of the respective contributions of each model to the shared variance between fMRI and MEG RDMs. The commonality (i.e. shared variance between model, fMRI and MEG RDM) for a model of interest (M1 in *eq. 5*) that is not explained by a second model (M2) is calculated as follows:

$$C(MEG,fMRI,M1) = R^2_{MEG,fMRI,M2} + R^2_{MEG,M1,M2} - R^2_{MEG,M2} - R^2_{MEG,fMRI,M1,M2} \tag{5}$$

(RDM elements that contain NaN values in any model are excluded from the analysis.)

Significance for these commonalities was determined using cluster-based permutation tests, following the same method as explained above for model-based MEG RSA. Further, significant clusters in *Figure 6E* were masked by only including spatiotemporal locations where both MEG-Model RDM correlation and fMRI-Model RDM correlation are significant. Lastly, we assessed the correspondence between group-averaged MEG and fMRI RDMs using Spearman correlation, resulting in a cross-modal RDM similarity time course for each ROI (*Figure 6F*, gray shading). Specifically, the squared Spearman rho was calculated and compared with the model-based commonality measures derived from *eq. 4* or *eq. 5*. The squared Spearman rho provides the upper bound for the maximal amount of shared variance between MEG and fMRI RDMs to be explained by the theoretical models (*Hebart et al., 2018*).

## Acknowledgements

This research was supported by the Intramural Research Program of the National Institutes of Health/National Institute of Neurological Disorders and Stroke, and National Science Foundation (BCS-1753218, to BJH). BJH further acknowledges support by Klingenstein-Simons Neuroscience Fellowship. CGG was supported by the Department of State Fulbright program. We thank Brian Maniscalco and Tom Holroyd for helpful discussions on code implementation and data acquisition, respectively.

## Additional information

### Funding

| Funder | Grant reference number | Author |
| --- | --- | --- |
| National Institute of Neurological Disorders and Stroke | | Biyu J He |

| Klingenstein-Simons Neuroscience Fellowship | | Biyu J He |
| U.S. Department of State | The Fulbright Program | Carlos González-García |
| National Science Foundation | BCS-1753218 | Biyu J He |

The funders had no role in study design, data collection and interpretation, or the decision to submit the work for publication.

## Author contributions

Matthew W Flounders, Conceptualization, Software, Formal analysis, Validation, Investigation, Visualization, Methodology, Writing—original draft; Carlos González-García, Formal analysis, Funding acquisition, Investigation, Visualization; Richard Hardstone, Software, Formal analysis, Validation, Visualization, Methodology, Writing—original draft; Biyu J He, Conceptualization, Resources, Data curation, Supervision, Funding acquisition, Validation, Methodology, Project administration, Writing—review and editing

## Author ORCIDs

Matthew W Flounders http://orcid.org/0000-0001-7014-4665
Carlos González-García http://orcid.org/0000-0001-6627-5777
Richard Hardstone http://orcid.org/0000-0002-7502-9145
Biyu J He http://orcid.org/0000-0003-1549-1351

## Ethics

Human subjects: The experiment was approved by the Institutional Review Board of the National Institute of Neurological Disorders and Stroke (under protocol #14-N-0002). All subjects provided written informed consent.

## Decision letter and Author response

Decision letter https://doi.org/10.7554/eLife.41861.014
Author response https://doi.org/10.7554/eLife.41861.015

## Additional files

### Supplementary files

• Source code 1. Source data and code for *Figures 4*, *5* and *6*.
DOI: https://doi.org/10.7554/eLife.41861.011
• Transparent reporting form
DOI: https://doi.org/10.7554/eLife.41861.012

### Data availability

All data generated or analysed during this study are included in the manuscript and supporting files.

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
