## [Decision Letter]

Thank you for sending your article entitled "Neural dynamics of visual ambiguity resolution by perceptual prior" for peer review at *eLife*. Your article has been reviewed by three peer reviewers, one of whom is a member of our Board of Reviewing Editors, and the evaluation has been overseen by Michael Frank as the Senior Editor.

The reviewers have indicated that the topic of the paper, namely identifying the temporal dynamics underlying the Mooney recognition effect and controlling for non-content-specific effects such as increased attention, salience or decreased task difficulty is very valuable and novel. However, the reviewers remain cautious about whether the data presented conclusively allows to address these claims.

1) The paper claims that the results are image-specific, but Figure 4D shows condition-specific results. In particular, Figure 4D only shows that recognition decreases between stimulus pattern dissimilarity. But this effect can be driven by many factors, e.g. decreased noise.

2) Figure 4E shows a significantly positive correlation between the representational similarity structure for recognized Mooney images and unambiguous images, but not that this effect is greater for post vs. pre Mooney images. Therefore, this effect cannot be conclusively related to Mooney image disambiguation.

*Reviewer #1:*

This paper is on perceptual processing with respect to prior knowledge. They use Mooney images, (binary images without recognizable content), which after one has seen the underlying grey scale image, are easily identified. This is a very powerful perceptual effect and allows the investigation of how prior information affects recognition. The paper uses MEG (and fMRI) in combination with a series of clever time resolved decoding approaches to show "time-courses of dissociation".

In addition they used representational similarity analysis (RSA) to show the time-course of similarities between pre and post (same physical stimulus) and post and gray (same percept that is recognition). Not too surprisingly, they show that these time resolved similarities differ, with the stimulus based similarities peaking early and the recognition based similarities peaking later.

Finally, they employ a powerful model based RSA approach where they investigate the commonalities of RSA based on MEG (as before), fMRI and a theoretically predicted RSA (i.e. a model). The model can incorporate recognition (high similarity between post and gray) etc. Importantly, by looking for commonalties across MEG and fMRI, they can, based on pattern similarity, fuse fMRI and MEG. Although clever and informative a similar approach has already been published (visual object recognition) (Cichy et al., 2014).

Although the presented data are very interesting and show what can be done with a clever multivariate methods, including model-based RSA analyses, the promise of the title "Neural dynamics of visual ambiguity resolution by perceptual prior" is not fulfilled by this paper. Potentially, this data could give us some insights on *how* the integration of prior and incoming visual information works. This is only vaguely addressed, e.g. by data shown in Figure 3C.

In addition, one could argue that the novelty of this paper is only incremental: In a previous paper by Cichy et al., 2014, and a subsequent paper by Hebart describing a similar approach using model based RSA (Hebart et al., 2018) similar results were obtained. They studied object recognition, which is also based on prior information (volunteers know the objects and have seen them before in a different manner), although there is no control condition (i.e. identical visual stimulus, but different percept) as in a Mooney faces experiment.

The current paper should either provide more information about the neural dynamics of visual ambiguity resolution or at least explain how their approach adds novel insights over and above the papers mentioned above.

*Reviewer #2:*

This study uses fMRI and MEG and an accomplished psychophysical paradigm to investigate how experience-driven image recognition affects neural responses – across time and space. To do so, it uses several multivariate analyses as well as multivariate MEG-fMRI data fusion.

The key findings are that (i) that experience-driven recognition can be decoded from 300ms onwards based on MEG response patterns, (ii) that this information (see i) persists in a stationary manner over time, (iii) that recognition increases MEG pattern distances between Mooney images, (iv) that the representational geometry (MEG-based) of recognized Mooney images correlates significantly with that for the corresponding original gray-scale images and (v) that shared variance between MEG and fMRI RDMs uniquely accounted for by a recognition model is widespread in the brain (in all ROIs) from 500ms onwards.

The methodology is without doubt advanced and the general question that this study is supposed to address is of wide general interest. However, my first main concern is that the authors do not introduce a specific hypothesis nor outline exactly how this research is going bring us closer to understanding how experience guides recognition. As a result, the study, although being informative, comes across as "fishing expedition".

Another major concern is that the authors do not report statistically solid univariate findings, which makes it impossible to relate the findings reported here to previous imaging studies employing a similar paradigm. The authors do present SVM weight-maps. However, these maps are anecdotal at best as they are not statistically evaluated in any way. Reporting univariate fMRI data would also be extremely valuable, as it would for example enable readers to assess how the MEG-fMRI modeling results relate to fMRI response amplitude (and SNR).

Furthermore, the authors make a claim that is not fully supported by their findings: they state that "This analysis showed that image-specific information for post-disambiguation Mooney images rises higher than their pre-disambiguation counterparts starting from ~500 ms" based on finding iii. This is misleading, because greater between-image pattern distances do not directly imply greater stimulus information. This finding could, for example, also be explained by noisier responses for recognized Mooney images.

Another issue is that a crucial test is missing related to finding iv (Figure 4E). This finding implies that recognizing Mooney images causes representational geometry (MEG based) to become more similar to that for the corresponding set of gray-scale images. However, the authors need to demonstrate that this increase in RDM-RDM similarity is significantly greater for the Post RDM as compared to the Pre RDM.

Finally, I don't see why finding ii is of interest (Figure 3D). To me it is unclear what sets this case of (MEG) pattern information persistence apart from previous reports if this phenomenon (e.g. Carlson et al., 2013), and how it functionally relates to experience-driven recognition.

Given these issues, I do not recommend publication of this manuscript in its current state.

*Reviewer #3:*

Summary:

Flounders and colleagues investigated how the visual and cognitive processing during recognition of images unfolds over time. To pinpoint the effects of the prior, including knowledge of the image content and expectation, they presented participants with two-tone "Mooney" images that are initially difficult to recognize but after disambiguation allow recognition of the stimulus (take the famous picture of the Dalmatian dog as an example). Using MEG decoding, the authors show striking differences between Mooney images before recognition ("pre") vs. after recognition ("post") emerging after ~300 ms. Using representational similarity analysis (RSA) to compare between different experimental phases, they show that, as expected, the similarity of stimulus-related patterns of activity increases rapidly after stimulus onset (comparing pre and post-recognition Mooney images), while recognition-related patterns emerge much later ~800 ms (comparing real images with post-recognition). By relating their results to a separate 7T fMRI dataset with model-based RSA, for several regions of interest (ROIs) the authors reveal time courses of information specific to different model components, related to stimulus-based, recognition-based and attention-based processes. They find that stimulus-based processes exhibit an early and a late peak, recognition-based processes dominate throughout time and region after ~500 ms, and attention-based processes are very specifically located to frontal and parietal ROIs at specific time points. They interpret these results in light of the effects of prior experience on object recognition.

Assessment:

The authors address a timely and interesting question of how prior experience affects visual and cognitive processing in the human brain. The manuscript uses state-of-the-art methodology in MEG decoding, RSA and MEG-fMRI fusion, and all statistical analyses appear to be sound. I specifically liked the combination of MEG and fMRI data for spatiotemporally-resolved analysis, and the related results are fascinating. In addition, I very much liked the addition of a control condition to make sure the results are not merely due to stimulus repetition (post-recognition images have been seen more frequently than pre-recognition stimuli) or simple stimulus-association effects.

At the same time, I believe the authors make some claims not supported by the data. They highlight that part of the novelty of their work has to do with the fact that previous work on this topic did not reveal image-specific results and, indeed, the authors do report image-specific findings in Figure 4E. However, in contrast to the authors' claim, the other effects using RSA are likely not stimulus-specific. For example, the results in Figure 4D are averaged across stimuli, leading to *condition*-specific effects. To achieve stimulus-specific effects, the authors would have to either identify the similarity for the same stimulus to itself or identify the difference between same stimulus and different stimulus within different periods of the experiment. They could do this by carrying out a split-half analysis and calculating the difference (within – between). This would be equivalent to a stimulus-specific decoding analysis. I think this kind of analysis would be useful to support their results. Alternatively, the authors may want to adjust this description of their results with respect to stimulus-specificity in the Materials and methods, Results, and Discussion.

A similar argument could be made regarding the model-based MEG-fMRI fusion results. The stimulus-specific model focuses on gross differences between Mooney images and greyscale images, rather than individual images. The recognition-specific model assumes that images post-recognition all become different from each other, which would lead to high dissimilarity. However, in line with the authors' interpretation of their prior work (Gonzalez-Garcia et al., 2018), one could also argue that they should become more similar to each other (when treated as the class of objects rather than individual images). In addition, their model interpretation would assume that the image itself should at least become more similar to itself, i.e. according to their interpretation, in my understanding the model would have to contain off-diagonal elements for the same image between grayscale and post-recognition periods.

To strengthen their conclusions, I would suggest the addition of stimulus-specific or at least category-specific (e.g. animate – inanimate) decoding analyses. Further, I would suggest carrying out a category-specific analysis (e.g. animate – inanimate) to confirm the claims that the results are indeed recognition-related.

While, as mentioned above, the addition of a control analysis is great, it only makes up a fraction of the other conditions. Therefore, the absence of decoding or RSA effects may be due to reduced power. What would the equivalent analysis look like for the experimental data if it is similarly reduced in size?

[Editors' note: further revisions were requested prior to acceptance, as described below.]

Thank you for resubmitting your work entitled "Neural dynamics of visual ambiguity resolution by perceptual prior" for further consideration at *eLife*. Your revised article has been favorably reviewed by three reviewers, one of whom is a member of our Board of Reviewing Editors, and the evaluation has been overseen by Michael Frank as the Senior Editor.

The manuscript has been improved but there are some remaining issues that need to be addressed before acceptance, as outlined below:

As you will see below there are remaining issues with Figure 4E as outlined in the comment by reviewer #2. The new Figure 5 needs further clarifications for the reader and there is one outstanding issue related to how exemplar i.e. image specific these findings are. Finally the model needs either adjustment or a justification in the Discussion section (see comment by reviewer #3).

*Reviewer #2:*

I am glad to see that the authors have found an elegant way of addressing my most important issue – the lack of direct support for the claim of recognition driven content-specific effects. I think that this issue is now covered by the results shown in Figure 5.

I still would like to insist on a revision of Figure 4E. The pink line in the figure is referred to as a "recognition-based" time course. This is misleading as the difference between the pink and the green line can be explained by differences in representational geometry due to recognition *and* due to differences related to presenting a Gray vs. Mooney images. Hence, there is no (significant) evidence for this effect being more pronounced for the Post-Gray than the Pre-Gray comparison. Therefore, the authors should explicitly clarify that the difference shown in this figure cannot be conclusively attributed to recognition. They could also simply omit 4E as the content specific recognition effect is now directly demonstrated in Figure 5. The authors could also consider to conceptually link Figure 4 and 5 by clarifying that Figure 4D leaves open the question if recognition-driven enhanced (and equal-to Gray) representational dissimilarity for Mooney images is driven by activation patterns to Mooney images becoming more similar to their Grayscale counterparts. At present, a compelling motivation for the Figure 5 analysis is missing at the start of the corresponding Results section.

With respect to my introduction-related comment, I still think that the clarity of the paper would be enhanced by including a more specific hypothesis. As noted in the rebuttal, this presently boils down to this sentence: "these observations raise the intriguing possibility that slow, long-distance recurrent neural dynamics involving large-scale brain networks are necessary for prior-experience-guided visual recognition." I find it hard to see how this rather broad and vague hypothesis is a natural motivation for the specific research preformed during this study nor how it is precisely addressed by the findings.

*Reviewer #3:*

I would like to thank the authors for taking up many of my suggestions. I believe the clarifications in the text and the addition of the novel stimulus-specific analyses greatly strengthened the paper and the conclusions that can be drawn from the results. Nevertheless, I have some remaining reservations.

It is correct that the analyses in Figure 4D were carried out at the image level. Nevertheless, in my understanding it is impossible to tell whether these effects are image-specific. This is a small correction to my original assessment where I said these results were condition-specific, while in fact they simply do not allow distinguishing between condition-specific and image-specific effects. The goal of this analysis is to show differences in the image-specific effects between the three conditions. Using the dissimilarity matrices in 4B and the analyses of the authors in 4C, then indeed the expected dissimilarity matrix for image-specific effects would exhibit low dissimilarity everywhere within a condition. This dissimilarity would then be expected to be different for each condition and change across time. This is the result the authors showed in 4D. However, for condition-specific effects the expected dissimilarity matrix would show low dissimilarity within condition, as well. The only way to tell these apart is by comparing the dissimilarity within image to the dissimilarity between. Since the authors already conducted this analysis, it is just a matter of clarification that the results in 4D cannot distinguish between condition-specific and image-specific results.

Regarding the MEG-fMRI fusion modeling, I agree with the authors that the model is a choice the author has to make, but it has to be both justified and consistent. For the former, perhaps it would make sense to spell out the construction of the models in some more detail. For the latter, if the authors do not want to make adjustments, I would suggest discussing those limitations. The "recognition" model assumes that (1) Mooney images in the Pre-phase are not recognized and are all similarly "unrecognized", (2) all recognized stimuli are different from each other, and (3) all recognized stimuli are as different from each other as they are from unrecognized images. I can follow (1). However, as I mentioned, for (2) according to the authors' interpretation the "diagonals of between-condition squares" should be similar to each other. Without this, the model is inconsistent. The authors argued they cannot predict the exact value expected for those cells. However, since they are using the Spearman correlation, they would just need to choose if the dissimilarity is lower than within Pre (i.e. the blue square), the same, or higher. If the authors cannot decide, they could leave out those cells and remain agnostic. Note that, however, the same issue of comparability arises regarding (3): it seems like an even stronger assumption that in terms of recognition all recognized images are as different from each other as they are from unrecognized images. The authors should either be explicit about this, adjust the model, or remove those cells from the analysis.

---

## [Author Response]

The reviewers have indicated that the topic of the paper, namely identifying the temporal dynamics underlying the Mooney recognition effect and controlling for non-content-specific effects such as increased attention, salience or decreased task difficulty is very valuable and novel. However, the reviewers remain cautious about whether the data presented conclusively allows to address these claims.1) The paper claims that the results are image-specific, but Figure 4D shows condition-specific results. In particular, Figure 4D only shows that recognition decreases between stimulus pattern dissimilarity. But this effect can be driven by many factors, e.g. decreased noise.

Firstly, because we showed that between-image dissimilarity is higher for post- than pre- images, we believe the above comment meant to say “Figure 4D only shows that recognition *increases* between stimulus pattern dissimilarity. But this effect can be driven by many factors, e.g. *increased* noise.”

Second, we would like to point out that for the analysis shown in Figure 4D, we had already conducted a control analysis using Euclidean distance and cross-validated Euclidean distance. The results of this analysis are presented in Figure 4—figure supplement 2, which may have been overlooked. Importantly, cross-validated Euclidean distance is a metric that is unaffected by noise in the data due to cross-validation (Guggenmos et al., 2018). Specifically, it is calculated as:

dEuclidean,c.v.2x,y=x-yA(x-y)[B]T, (1) where A and B denote the two partitions of the data within each cross-validation fold. (For this analysis, we used a 3-fold cross-validation scheme.) This way, noise in the data cancels out, and the cross-validated Euclidean distance is only driven by signal, that is, the component of the data that is consistent across partitions. As explained by Guggenmos et al., this approach “can improve the reliability of distance estimates when noise levels differ between measurements” (P. 438).

As expected, this control analysis using cross-validated Euclidean distance yielded very similar result to that shown in Figure 4D. The method for this analysis was previously tucked away in the legend to Figure 4—figure supplement 2. We have now included detailed method in the manuscript (Materials and methods, subsection “Euclidean distance”) and have explained the rationale and interpretation of this control analysis better in Results (subsection “Disambiguation increases across-image dissimilarity in neural dynamics”).

Third, to further boost the claim for content-specific effects, we have now conducted a stimulus-specific decoding analysis using the “correlation (within-between)” metric as suggested by reviewer #4. This analysis allowed us to test, at the single-trial level, how well we can distinguish neural activities elicited by a Mooney image presented before vs. after disambiguation, and how well we can distinguish a pre- or post-disambiguation Mooney image from its corresponding Grayscale image. We were very pleased by the clarity and the robustness of this finding, and are deeply grateful to reviewer #4 for suggesting this analysis. The results, presented in a newly added figure (Figure 5), show that within a ~300 ms window after stimulus onset, a Mooney image – whether presented before or after disambiguation – is well separable at the single-trial level from its corresponding grayscale image. By contrast, after 500 ms post-stimulus-onset, a post-disambiguation Mooney image is entirely indistinguishable from its matching grayscale image at the single trial level despite differences in stimulus features (but well separable from other grayscale images, Figure 5D), while a pre-disambiguation Mooney image is well separable from the same image presented after disambiguation or its matching grayscale image (Figure 5C). These results reveal an image-specific shift in neural representation toward the relevant prior experience.

In a complementary analysis, we quantified the strength of the diagonals of between-condition squares (referred to as “off-diagonal elements for the same image” by reviewer #4) as compared to the off-diagonal elements in the same between-condition squares, again using the “correlation (within-between)” measure (Figure 5D). This analysis quantifies the similarity between the neural representation of the same/matching image in different perceptual conditions above and beyond its similarity to other images (e.g., is post-Image-A represented more similarly to pre-Image-A than to pre-Image-B)? The results, presented in Figure 5D, show that neural activities reflect similarities of stimulus features in an early (<500 ms) time window and similarities of recognition content in a late (>500 ms) time period.

We would like to refer the editors and reviewers to the newly added text in the Results (subsection “Comparing image-specific dynamic neural representations across perceptual conditions at the single-trial level”), Discussion (fifth paragraph), Materials and methods (subsection “Single-trial separability”), and the new Figure 5 for further details. We believe that these new observations provide strong evidence for content-specific neural effects – encoding of stimulus input in the early (<300 ms) time period and of recognition content in the late (>500 ms) time period, which further strengthens our main conclusions.

2) Figure 4E shows a significantly positive correlation between the representational similarity structure for recognized Mooney images and unambiguous images, but not that this effect is greater for post vs. pre Mooney images. Therefore, this effect cannot be conclusively related to Mooney image disambiguation.

In the analysis shown in Figure 4E, we had indeed probed the correlation between the representational geometry of pre-disambiguation Mooney images and grayscale images. As we mentioned in the text, “Correlations of Pre-Pre and Gray-Gray squares of the RDM were not significant at any time point.” Since no significance was found, we chose not to include this trace in Figure 4E to avoid cluttering the figure (the full figure is included as Author response image 1). This result contrasts with the correlations between Post-Post and Gray-Gray squares, which yielded sustained significant (*p* < 0.05, cluster-based permutation test) clusters after 500 ms (Figure 4E, magenta). Likely due to insufficient statistical power, a direct contrast between r(Pre-Pre, Gray-Gray) and r(Post-Post, Gray-Gray) did not yield significant clusters following correction for multiple comparisons using a cluster-based permutation test. Nonetheless, we believe that this analysis is valuable, given the sustained significance in the late (>500 ms) time period for the recognition-based representation (correlating representational geometry between post-Mooney and grayscale images), and the lack of significance for the control analysis (correlating representational geometry between pre-Mooney and grayscale images).

**Author response image 1. respfig1:** Same as Figure 4E, now showing element-wise correlations between the Pre-Pre and Gray-Gray triangles in the RDM as the black dashed line. No significant cluster was found for this comparison at a level of *p* < 0.05 (cluster-based permutation test).

In sum, we believe that our results, including the new addition, present strong evidence for content-specific neural effects related to stimulus processing and subjective recognition during prior-guided visual perception, and reveal their respective time courses.

Reviewer #1:

This paper is on perceptual processing with respect to prior knowledge. They use Mooney images, (binary images without recognizable content), which after one has seen the underlying grey scale image, are easily identified. This is a very powerful perceptual effect and allows the investigation of how prior information affects recognition. The paper uses MEG (and fMRI) in combination with a series of clever time resolved decoding approaches to show "time-courses of dissociation".In addition they used representational similarity analysis (RSA) to show the time-course of similarities between pre and post (same physical stimulus) and post and gray (same percept i.e. recognition). Not too surprisingly, they show that these time resolved similarities differ, with the stimulus based similarities peaking early and the recognition based similarities peaking later.Finally, they employ a powerful model based RSA approach where they investigate the commonalities of RSA based on MEG (as before), fMRI and a theoretically predicted RSA (i.e. a model). The model can incorporate recognition (high similarity between post and gray) etc. Importantly, by looking for commonalties across MEG and fMRI, they can, based on pattern similarity, fuse fMRI and MEG. Although clever and informative a similar approach has already been published (visual object recognition) (Cichy et al., 2014).

The contribution of the present study is not in methodology development, but in using recently developed methods for probing multivariate neural representations in dynamic, whole-head MEG signals and merging neural data across modalities at an informational level (e.g., Kriegeskorte et al., 2008; Cichy et al., 2014; Hebart et al., 2018; Guggenmos et al., 2018) to reveal neural mechanisms underlying prior knowledge’s influence on visual perception and recognition. Previous studies applying these methods have used images depicting clear, high-contrast, isolated objects, where the prior knowledge invoked by the images (e.g., the knowledge of cows as an animal category) was solidified typically decades ago during development. By contrast, our paradigm allows the establishment of a prior knowledge de novo in an extremely fast and robust manner; this allows us to probe and contrast visual perception/recognition without vs. with prior knowledge and reveal the time courses of the involved neural computations. The distinction of our results from previous findings is also underscored by different latencies of the identified neural effects: we find recognition-related neural effects with temporal latencies (>500 ms following stimulus onset) much later than most previously reported neural effects related to object recognition using MEG (typically within 500 ms). This difference and the related considerations were addressed in the Introduction (third paragraph) and Discussion (sixth paragraph). We now better explain our topic of investigation and its broader significance in the opening paragraph of Discussion:

“Despite the pervasive need to resolve stimulus ambiguity (caused by occlusion, clutter, shading, and inherent complexities of natural objects) in natural vision (Olshausen and Field, 2005) and the enormous power that prior knowledge acquired through past experiences wields in shaping perception (Helmholtz, 1924; Albright, 2012), the neural mechanisms underlying prior-guided visual recognition remain mysterious. Here, we exploited a dramatic visual phenomenon, where a single exposure to a clear, unambiguous image greatly facilitates recognition of a related degraded image, to shed light on dynamical neural mechanisms that allow past experiences to guide recognition of impoverished sensory input.”

Although the presented data are very interesting and show what can be done with a clever multivariate methods, including model-based RSA analyses, the promise of the title "Neural dynamics of visual ambiguity resolution by perceptual prior" is not fulfilled by this paper. Potentially, this data could give us some insights on how the integration of prior and incoming visual information works. This is only vaguely addressed, e.g. by data shown in Figure 3C.In addition, one could argue that the novelty of this paper is only incremental: In a previous paper by Cichy et al., 2014, and a subsequent paper by Hebart describing a similar approach using model based RSA (Hebart et al., 2018) similar results were obtained. They studied object recognition, which is also based on prior information (volunteers know the objects and have seen them before in a different manner), although there is no control condition (i.e. identical visual stimulus, but different percept) as in a Mooney faces experiment.The current paper should either provide more information about the neural dynamics of visual ambiguity resolution or at least explain how their approach adds novel insights over and above the papers mentioned above.

We hope that the above responses to the reviewer’s overall assessment and the editors’ comments have sufficiently addressed these concerns.

Reviewer #2:

[…] The methodology is without doubt advanced and the general question that this study is supposed to address is of wide general interest. However, my first main concern is that the authors do not introduce a specific hypothesis nor outline exactly how this research is going bring us closer to understanding how experience guides recognition. As a result, the study, although being informative, comes across as "fishing expedition".

We respectfully disagree with this characterization of our study. As we presented in Introduction, the current study indeed tests a specific hypothesis:

“Previous neuroimaging studies have observed that disambiguation of Mooney images induces widespread activation and enhanced image-specific information in both visual and frontoparietal cortices. […] Together with a recent finding of altered content-specific neural representations in frontoparietal regions following Mooney image disambiguation, these observations raise the intriguing possibility that slow, long-distance recurrent neural dynamics involving large-scale brain networks are necessary for prior-experience-guided visual recognition.”

Another major concern is that the authors do not report statistically solid univariate findings, which makes it impossible to relate the findings reported here to previous imaging studies employing a similar paradigm. The authors do present SVM weight-maps. However, these maps are anecdotal at best as they are not statistically evaluated in any way. Reporting univariate fMRI data would also be extremely valuable, as it would for example enable readers to assess how the MEG-fMRI modeling results relate to fMRI response amplitude (and SNR).

Statistically solid univariate findings using the fMRI data set have already been reported in a previous publication (Gonzalez-Garcia et al., 2018). Since the strength of MEG is in the temporal domain, not the spatial domain, and due to volume conduction, we believe that a massive univariate analysis using the MEG data set in the context of the present study would be superfluous. This is also not in line with most recent studies employing multivariate analyses applied to MEG data (e.g., Carlson et al., 2013; Cichy et al., 2014; Hebart et al. 2018, referenced by reviewers #1 and #2).

Furthermore, the authors make a claim that is not fully supported by their findings: they state that "This analysis showed that image-specific information for post-disambiguation Mooney images rises higher than their pre-disambiguation counterparts starting from ~500 ms" based on finding iii. This is misleading, because greater between-image pattern distances do not directly imply greater stimulus information. This finding could, for example, also be explained by noisier responses for recognized Mooney images.

Please see our response to the editor’s point #1 above.

Another issue is that a crucial test is missing related to finding iv (Figure 4E). This finding implies that recognizing Mooney images causes representational geometry (MEG based) to become more similar to that for the corresponding set of gray-scale images. However, the authors need to demonstrate that this increase in RDM-RDM similarity is significantly greater for the Post RDM as compared to the Pre RDM.

Please see our response to the editor’s point #2 above.

Finally, I don't see why finding ii is of interest (Figure 3D). To me it is unclear what sets this case of (MEG) pattern information persistence apart from previous reports if this phenomenon (e.g. Carlson et al., 2013), and how it functionally relates to experience-driven recognition.

Carlson et al. 2013 is no doubt a classic in the literature. However, it addresses a distinct question from the current study: dynamical neural mechanisms underlying recognition of clear, high-contrast color images of isolated objects (Carlson study) vs. dynamical neural mechanisms underlying experience-guided recognition of degraded, black-and-white images of objects embedded in scenes, where recognition without prior experience is extremely difficult (our study). In addition, the cross-decoding result in Carlson et al. (Figure 6A therein) did not show “pattern information persistence”, but rather transient effects that were fast changing over time – as shown by the diagonal pattern of significant decoding which contrasts with the rectangular pattern in our Figure 3D. (But again, these two analyses are asking very different questions in two studies that have different aims.)

What this analysis (Figure 3D) shows is that neural activity patterns distinguishing perceptual stage (pre- vs. post-disambiguation) are relatively sustained over time. Although this analysis is not content-specific (as we clearly acknowledge in the manuscript), it sets up the stage for the content-specific analyses and results presented thereafter.

Given these issues, I do not recommend publication of this manuscript in its current state.

We hope that we have satisfactorily addressed the reviewer’s concerns.

Reviewer #3:

[…] At the same time, I believe the authors make some claims not supported by the data. They highlight that part of the novelty of their work has to do with the fact that previous work on this topic did not reveal image-specific results and, indeed, the authors do report image-specific findings in Figure 4E. However, in contrast to the authors' claim, the other effects using RSA are likely not stimulus-specific. For example, the results in Figure 4D are averaged across stimuli, leading to condition-specific effects. To achieve stimulus-specific effects, the authors would have to either identify the similarity for the same stimulus to itself or identify the difference between same stimulus and different stimulus within different periods of the experiment. They could do this by carrying out a split-half analysis and calculating the difference (within – between). This would be equivalent to a stimulus-specific decoding analysis. I think this kind of analysis would be useful to support their results. Alternatively, the authors may want to adjust this description of their results with respect to stimulus-specificity in the Materials and methods, Results, and Discussion.

Please see our response to the editors’ point #1 above. We are deeply grateful to the reviewer for suggesting the split-half analysis using “correlation (within-between)” metric, the results of which are included in the newly added Figure 5 and described in a newly added Results subsection “Comparing image-specific dynamic neural representations across perceptual conditions at the single-trial level”). Related methods are described in Materials and methods, subsection “Single-trial separability”. Although we did not use a split-half analysis exactly, we performed such an analysis at the single-trial level to calculate the “correlation (within – between)” metric.

We also note that we think the results in Figure 4D were indeed image-specific instead of condition-specific. This is because the time courses in Figure 4D were averaged across individual image-pairs within each perceptual condition, where each value quantifies the dissimilarity between neural activity patterns related to those two individual images. Thus, this analysis shows that neural activity patterns elicited by post-disambiguation Mooney images are more distinct from each other than those elicited by the same images presented pre-disambiguation. In a control analysis (shown in Figure 4—figure supplement 2, now better explained in Results, subsection “Disambiguation increases across-image dissimilarity in neural dynamics”, and Materials and methods, subsection “Euclidean distance”), the results in Figure 4D were reproduced using cross-validated Euclidean distance, which is only contributed by signal components that are consistent across separate partitions of the data (hence, suppressing the contribution of random noise). Nonetheless, since this analysis was not conducted at the level of single trials, we have now removed the term “image-specific information” when describing this analysis (in the Abstract, title and concluding paragraph in the corresponding Results section).

A similar argument could be made regarding the model-based MEG-fMRI fusion results. The stimulus-specific model focuses on gross differences between Mooney images and greyscale images, rather than individual images. The recognition-specific model assumes that images post-recognition all become different from each other, which would lead to high dissimilarity. However, in line with the authors' interpretation of their prior work (Gonzalez-Garcia et al., 2018), one could also argue that they should become more similar to each other (when treated as the class of objects rather than individual images). In addition, their model interpretation would assume that the image itself should at least become more similar to itself, i.e. according to their interpretation, in my understanding the model would have to contain off-diagonal elements for the same image between grayscale and post-recognition periods.

We do not understand the reviewer’s comment “in line with the authors' interpretation of their prior work (Gonzalez-Garcia et al., 2018), one could also argue that they should become more similar to each other (when treated as the class of objects rather than individual images).” In this previous paper, we showed that dissimilarity between neural representation of individual images increases substantially after disambiguation, i.e., they became more different from each other. This was shown in several ways in that paper: i) overall redder hues of the Post-Post square than the Pre-Pre square of the RDMs (Figure 3B and 4A); ii) larger distances between dots representing individual images in the Post stage than the Pre stage in the multidimensional scaling (MDS) plots (Figure 3C and 4B); iii) statistical summary results (Figure 3D and 4C, cyan brackets).

The “off-diagonal elements for the same image between grayscale and post-recognition periods” (i.e., “the diagonal elements of between-condition squares” in our terminology, which we think is more accurate) have been systematically and comprehensively probed in the analysis described in the new Figure 5 (see our reply to the previous comment). This analysis quantifies these diagonals and compares them between each other (Figure 5C) and to the off-diagonal elements in the same between-condition squares (Figure 5D) using the empirical MEG data. We think this is a superior approach to including such diagonals in the between-condition squares of the model RDM. This is because the models were designed to probe relatively coarse effects (as we acknowledge in the manuscript), and it would be hard to know what arbitrary value to set such between-condition diagonals within each model, which contained binary values capturing coarser effects. All models are intended to capture certain aspects in the data (“all models are wrong, some are useful”). In fact, we think that it is wonderful (but not an a priori given) that this model-based fMRI-MEG fusion analysis probing relatively coarse effects yielded stimulus- and recognition-related neural activity time courses that are very much consistent with the earlier content-specific analyses applied to the MEG data alone (Figures 4 and 5). But, of course, the model-based fusion analysis provided further insight into the spatial dimension by bringing in the fMRI data.

To strengthen their conclusions, I would suggest the addition of stimulus-specific or at least category-specific (e.g. animate – inanimate) decoding analyses. Further, I would suggest carrying out a category-specific analysis (e.g. animate – inanimate) to confirm the claims that the results are indeed recognition-related.

We hope that we have satisfactorily addressed the concerns regarding stimulus-specific effects (summarized in our response to the editors’ point #1). We strongly believe that the current results provide very clear and very robust content-specific recognition-related effects: for example, we show that a post-disambiguation Mooney image is indistinguishable at the single-trial level from its matching grayscale image from ~500 ms onward (Figure 5C, dark blue) but well separable from other grayscale images (Figure 5D, dark blue), while in the same time period a pre-disambiguation Mooney image is well separable at the single-trial level from the same image presented post-disambiguation or from its matching grayscale image (Figure 5C, orange and green). These results reveal an image-specific shift in neural representation toward the relevant prior experience that guides perception. Furthermore, our previously included control analysis for Figure 4D, using cross-validated Euclidean distance (Figure 4—figure supplement 2), demonstrated that the increase in between-image dissimilarity following disambiguation was not driven by changing levels of noise in the data.

Given that we have shown clear content-specific effects at the level of individual images, we respectfully think that a category-level analysis is beyond the scope of this study.

While, as mentioned above, the addition of a control analysis is great, it only makes up a fraction of the other conditions. Therefore, the absence of decoding or RSA effects may be due to reduced power. What would the equivalent analysis look like for the experimental data if it is similarly reduced in size?

We have performed a control analysis for Figure 3A-B which matches the statistical power between real and catch image sets. Since there were only 6 catch image sets, we randomly selected 6 real image sets and re-conducted decoding of presentation stage (Pre- vs. Post-disambiguation). After matching the statistical power of catch image sets, we still obtained significant decoding of presentation stage using the SCP band, as shown in Author response image 2. Qualitatively similar results were obtained using the ERF band (Author response image 2). Due to computational intensiveness, we did not perform cluster-based permutation test for this analysis.

**Author response image 2. respfig2:** Same as Figure 3A-B (black and green traces), except that 6 randomly selected real image sets were used to match the statistical power of catch image sets. Results from 10 such randomly selected subsets were averaged together, and mean and s.e.m. of decoding accuracy across subjects are plotted for both real (green) and catch (black) image sets. Horizontal bar indicates significant difference from chance level (*p* < 0.05, FDR corrected). Catch results are identical as in Figure 3A-B.

Since changes in statistical power would only systematically affect decoding accuracy (such as in Figure 3A-B), but not estimation of the mean (such as in Figure 4D and Figure 4—figure supplement 1), we did not perform a similar control analysis for Figure 4D. In other words, if we select 6 random real image sets and re-compute Figure 4D, we would not expect any systematic change in the result.

[Editors' note: further revisions were requested prior to acceptance, as described below.]

The manuscript has been improved but there are some remaining issues that need to be addressed before acceptance, as outlined below: As you will see below there are remaining issues with Figure 4E as outlined in the comment by reviewer #2. The new Figure 5 needs further clarifications for the reader and there is one outstanding issue related to how exemplar i.e. image specific these findings are. Finally the model needs either adjustment or a justification in the Discussion section (see comment by reviewer #3).

We are grateful to the editors and reviewers for the favorable evaluation of our previous revision and the additional helpful suggestions. We have thoroughly further revised the manuscript in line with the editors’ and reviewers’ comments. Major changes include:

- We have significantly toned down the interpretations and conclusions derived from Figure 4E, and now specifically state that this analysis only provides qualitative evidence (albeit from a very unique angle and yielding findings that are consistent with all the other analyses), which is quantitatively assessed by the ensuing model-driven data fusion analysis.

- In response to both reviewers’ comments suggesting that Figure 4D leads naturally to Figure 5, we have now swapped the order of the Results sections related to Figure 4E and Figure 5, such that the revised text describes results in the following sequence: Figure 4D → Figure 5 → Figure 4E → Figure 6. Although unconventional in terms of figure sequence, we think that this order fits better with the logical flow of the analyses, such that the questions opened up by Figure 4D are answered by Figure 5, and the qualitative evidence provided by Figure 4E is strengthened by Figure 6.

- We have revised the models in line with reviewer #4’s suggestions. The results obtained with these updated models are consistent with our previous findings but show stronger neural effects. As a result, Figure 6 as well as Figure 6—figure supplement 1 have been updated, and the related text has been thoroughly revised.

We believe that our revision has fully addressed all of the editors’ and reviewers’ remaining concerns and the manuscript has been further strengthened as a result. Please find below a point-by-point response to the editors’ and reviewers’ comments.

Reviewer #2:

I am glad to see that the authors have found an elegant way of addressing my most important issue – the lack of direct support for the claim of recognition driven content-specific effects. I think that this issue is now covered by the results shown in Figure 5.

We are pleased that the reviewer found the analysis reported in Figure 5 satisfactory.

I still would like to insist on a revision of Figure 4E. The pink line in the figure is referred to as a "recognition-based" time course. This is misleading as the difference between the pink and the green line can be explained by differences in representational geometry due to recognition and due to differences related to presenting a Gray vs. Mooney images. Hence, there is no (significant) evidence for this effect being more pronounced for the Post-Gray than the Pre-Gray comparison. Therefore, the authors should explicitly clarify that the difference shown in this figure cannot be conclusively attributed to recognition. They could also simply omit 4E as the content specific recognition effect is now directly demonstrated in Figure 5. The authors could also consider to conceptually link Figure 4 and 5 by clarifying that Figure 4D leaves open the question if recognition-driven enhanced (and equal-to Gray) representational dissimilarity for Mooney images is driven by activation patterns to Mooney images becoming more similar to their Grayscale counterparts. At present, a compelling motivation for the Figure 5 analysis is missing at the start of the corresponding Results section.

We have now significantly toned down and qualified the interpretations of Figure 4E, and present it as providing *qualitative* evidence consistent with the other analyses. Below we reproduce the most relevant text:

“As a control measure, the correlation between Pre-Pre and Gray-Gray squares of the RDM was not significant at any time point, suggesting that, as expected, representational geometry is different between conditions with different stimulus input and different recognition outcomes. […] In the final analysis presented below, we will quantitatively test this possibility using a model-driven MEG-fMRI fusion analysis that simultaneously elucidates the spatial dimension of the evolving neural dynamics.”

As mentioned above, we have also moved the Results section related to Figure 4E later, to provide a more smooth and direct transition between Figure 4D and Figure 5:

“This dramatic effect raises two important questions: 1) Is this effect driven by the neural representations of Mooney images shifting towards those of their respective grayscale counterparts? […] To answer these questions, we next probe how neural representation for a particular Mooney image changes following disambiguation at the single-trial level.”

We believe that Figure 4E is a valuable analysis to retain in the manuscript, since it is the only analysis that probes how the fine-grained representational geometry (the set of representational distances across all image pairs) compares between conditions. It shows that the representational geometry is significantly similar between Pre and Post conditions in an early time window (<300 ms), and between Post and Gray conditions in a late time window (>600 ms). In addition, the control analysis of comparing between Pre and Gray conditions did not yield any significant time point, as expected. We acknowledge that the result is not as strong as one would like, since a direct contrast between Post-Gray and Pre-Gray comparisons did not yield significance after correcting for multiple comparisons using cluster-based permutation test. Given these considerations, we have opted to leave the analysis in and present it as qualitative evidence.

With respect to my introduction-related comment, I still think that the clarity of the paper would be enhanced by including a more specific hypothesis. As noted in the rebuttal, this presently boils down to this sentence: "these observations raise the intriguing possibility that slow, long-distance recurrent neural dynamics involving large-scale brain networks are necessary for prior-experience-guided visual recognition." I find it hard to see how this rather broad and vague hypothesis is a natural motivation for the specific research preformed during this study nor how it is precisely addressed by the findings.

In the sentence quoted by the reviewer, we have now clarified that “slow” refers to “taking longer than 500 ms”. We believe that this is actually a very specific hypothesis, given that previous studies have typically reported recognition-related neural activity that concludes within 500 ms, as we previously stated in the same paragraph (reproduced below):

“By contrast, neural dynamics underlying recognition of intact, unambiguous images, as well as scene-facilitation of object recognition, typically conclude within 500 ms (Carlson et al., 2013; van de Nieuwenhuijzen et al., 2013; Kaiser et al., 2016; Brandman and Peelen, 2017). Together with a recent finding of altered content-specific neural representations in frontoparietal regions following Mooney image disambiguation (Gonzalez-Garcia et al., 2018), these observations raise the intriguing possibility that slow (taking longer than 500 ms), long-distance recurrent neural dynamics involving large-scale brain networks are necessary for prior-experience-guided visual recognition.”

We also note that not all valuable research derives from testing specific hypotheses and data-driven analyses are equally important for uncovering behaviorally relevant patterns in large, complex neural data sets without a priori biases. Some of our analyses may lie between hypothesis-driven and data-driven extremes, as we note in Introduction:

“To unravel neural mechanisms underlying prior experience’s influence on perception, an important unanswered question is how different information processing stages are dynamically encoded in neural activities.”

Reviewer #3:

I would like to thank the authors for taking up many of my suggestions. I believe the clarifications in the text and the addition of the novel stimulus-specific analyses greatly strengthened the paper and the conclusions that can be drawn from the results. Nevertheless, I have some remaining reservations.It is correct that the analyses in Figure 4D were carried out at the image level. Nevertheless, in my understanding it is impossible to tell whether these effects are image-specific. This is a small correction to my original assessment where I said these results were condition-specific, while in fact they simply do not allow distinguishing between condition-specific and image-specific effects. The goal of this analysis is to show differences in the image-specific effects between the three conditions. Using the dissimilarity matrices in 4B and the analyses of the authors in 4C, then indeed the expected dissimilarity matrix for image-specific effects would exhibit low dissimilarity everywhere within a condition. This dissimilarity would then be expected to be different for each condition and change across time. This is the result the authors showed in 4D. However, for condition-specific effects the expected dissimilarity matrix would show low dissimilarity within condition, as well. The only way to tell these apart is by comparing the dissimilarity within image to the dissimilarity between. Since the authors already conducted this analysis, it is just a matter of clarification that the results in 4D cannot distinguish between condition-specific and image-specific results.

We agree. As mentioned above, we have now re-ordered the text sections related to Figure 4E and Figure 5, such that the presentation of Figure 4D is followed by Figure 5 in the Results section. We have further added a paragraph at the end of Figure 4D section (subsection “Disambiguation increases across-image dissimilarity in neural dynamics”) to discuss the limitations of this analysis and provide a better transition to the analysis presented in Figure 5.

Regarding the MEG-fMRI fusion modeling, I agree with the authors that the model is a choice the author has to make, but it has to be both justified and consistent. For the former, perhaps it would make sense to spell out the construction of the models in some more detail. For the latter, if the authors do not want to make adjustments, I would suggest discussing those limitations. The "recognition" model assumes that (1) Mooney images in the Pre-phase are not recognized and are all similarly "unrecognized", (2) all recognized stimuli are different from each other, and (3) all recognized stimuli are as different from each other as they are from unrecognized images. I can follow (1). However, as I mentioned, for (2) according to the authors' interpretation the "diagonals of between-condition squares" should be similar to each other. Without this, the model is inconsistent. The authors argued they cannot predict the exact value expected for those cells. However, since they are using the Spearman correlation, they would just need to choose if the dissimilarity is lower than within Pre (i.e. the blue square), the same, or higher. If the authors cannot decide, they could leave out those cells and remain agnostic. Note that, however, the same issue of comparability arises regarding (3): it seems like an even stronger assumption that in terms of recognition all recognized images are as different from each other as they are from unrecognized images. The authors should either be explicit about this, adjust the model, or remove those cells from the analysis.

We would like to thank the reviewer for the very helpful suggestion. Both the consideration about Spearman correlation (only the ordering of values, not their absolute values, matters) and the strategy of excluding cells where the model is agnostic were great suggestions. We have now updated both the Stimulus and the Recognition model to address the points raised by the reviewer (specifically, 2 and 3). The new results are qualitatively consistent with what we presented in the previous submission, but show stronger neural effects related to stimulus processing. Given the extensiveness of changes to the text, we do not reproduce the revised text here, but would like to refer the reviewer to the relevant Results subsection “Model-driven MEG-fMRI data fusion spatiotemporally resolves neural dynamics related to stimulus, attention, and recognition processing”, as well as the revised Figure 6.